
**Fine-scale vertical structure of sound scattering layers over an east**
**border upwelling system and its relationship to pelagic habitat**
**characteristics**
Ndague DIOGOUL[1,6,*], Patrice BREHMER[2,3,5], Yannick PERROT[3], Maik TIEDEMANN[4],
Abou THIAM[1], Salaheddine EL AYOUBI[5], Anne MOUGET[3], Chloé MIGAYROU[3], Oumar
SADIO[2] and Abdoulaye SARRÉ[6]
[1] University Cheikh Anta Diop UCAD, Institute of Environmental Science (ISE), BP 5005
Dakar, Senegal
[2] IRD, Univ Brest, CNRS, Ifremer, LEMAR, Campus UCAD-IRD de Hann, Dakar, Senegal
[3] IRD, Univ Brest, CNRS, Ifremer, LEMAR, DR Ouest, Plouzané, France
[4] Institute of Marine Research IMR, Pelagic Fish, PO Box 1870 Nordnes, 5817 Bergen, Norway
[5] Institut National de Recherche Halieutique INRH, Agadir, Morocco
[6] Institut Sénégalais de Recherches agricoles ISRA, Centre de Recherches Océanographiques
de Dakar-Thiaroye (CRODT), BP 2221 Dakar, Senegal
*Corresponding author: diogoulndague@yahoo.fr
**Abstract**

22       Sound scattering layers 'SSLs' distribution and their relationship to pelagic habitat

characteristics is a first step to understand their role in ecosystem dynamics and their
interactions with other pelagic components such as small pelagic fish. In this study two areas
of the Senegalese shelf have been characterized during upwelling season corresponding to a
cold inshore area and a deeper and warmer stratified offshore area that was sharply separated
by a strong thermal boundary. Marine pelagic organisms usually aggregates and occurs as
'SSLs' on echosounder. Mean SSL thickness and SSL vertical depth increase with continental
shelf depth; thickest and deepest SSLs were observed in the offshore part of the continental
shelf. SSLs preferendum was reported for stratified water conditions rather than fresh upwelled
water. The SSLs spatiotemporal variability was mostly explained by bottom depth which
influence depth, thickness, and biomass of the SSLs. Diel period and water physico-chemical
characteristics had also an effect on SSL depth and SSL thickness but not on SSL biomass.
Despite chlorophyll-*a* has statistically no effect on SSLs structure, we report that the





chlorophyll-*a* peak was always located above or in the middle of the SSLs, often matching with
the peak of SSLs biomass. Such observations indicate trophic relationships, highlighting SSLs
being mainly composed of phytoplanktivorous organisms. Acoustic mapping technique of
mixed layer depth is not always efficient in east border upwelling system. Lastly, over the
Senegalese continental shelf the level of dissolved oxygen was not found limitative (no
hypoxia) for SSLs marine pelagic organisms during upwelling event.
**Keywords**: pelagic organism, micronekton, plankton, diel vertical migration (DVM), Senegal,
West Africa.

## 1    Introduction

Aggregations of marine pelagic organisms in ocean water columns can be observed

acoustically as sound-scattering layers (SSLs) (Evans and Hopkins, 1981; Cascão et al., 2017).
These SSLs represent the entire mid trophic level (Behagle et al., 2017) that comprise millions
of tonnes of zooplankton and micronekton, which are fundamental to ecosystem functioning,
particularly in productive upwelling areas (*e.g*., off the south coast of Senegal) (Auger et al.,
2016). Knowledge of the vertical structure of SSLs allows one to understand their role in
ecosystems better, information that can be used to monitor major environmental change and
variability. Most zooplankton and micronektonic taxa undergo diel vertical migration (DVM),
meaning that they reside in deep waters during the day and migrate toward the surface at night
to feed (Bianchi et al., 2013; Lehodey et al., 2015). DVM behaviors are influenced by
environmental cues (*e.g*., light, nutrients, and temperature) and predator-prey interactions
(Clark and Levy, 1988; Lampert, 1989). Thus, DVMs represent an essential biological process
in the ocean, one that also regulates the biological carbon pump (Hidaka et al., 2001).

The distribution of SSLs is influenced by a variety of environmental factors (Aoki and

Inagaki, 1992; Baussant et al., 1992; Dekshenieks et al., 2001; Marchal et al., 1993). Changes
in the structure and density of SSLs are associated with frontal zones (Aoki and Inagaki, 1992;
Baussant et al., 1992; Boersch-Supan et al., 2017; Coyle and Cooney, 1993). Oceanic front is a
relatively narrow zone of enhanced horizontal gradients of physical, chemical and biological
properties (temperature, salinity, nutrients etc.) that separates broader areas of different vertical
structure (stratification) (Belkin et al., 2009). Upwelling fronts are by now a very well
recognized part of the coastal upwelling process. Examples occur in many well studied systems,
including the upwelling off southern Senegal, south of Cap-Vert Peninsula (14.6°–13.5° North,



16.9°–17.6° West). The main originality of this upwelling system  compared to other upwelling
stems from its continental shelf that is broad and shallow, *i.e.*, 20–30 m over tens of kilometres
(Ndoye et al., 2017). Local bottom relief combined with the wind-induced upwelling establish
a typical upwelling that appear as a cold-water tongue. This cold-water tongue separates the
nutrient-poor warm offshore cell with a cold nutrient-rich coastal cell functioning as a retention
zone (Roy, 1998; Tiedemann and Brehmer, 2017). Physical variability for example off the
coastal shelf off Senegal (Capet et al., 2016; Ndoye et al., 2017) can impact marine pelagic
organisms at the individual or community level and such effects can be direct (*e.g.*, via
advection) or indirect (*e.g.*, via phytoplankton production fertilized by upwelled nutrients
(Urmy and Horne, 2016). Indeed, changes in water physicochemical properties and biological
activities induced by upwelling plays a structuring role on SSLs distribution. SSLs position is
often reported below the thermocline suggesting that temperature controls the SSLs vertical
distribution (Aoki and Inagaki, 1992; Baussant et al., 1992; Boersch-Supan et al., 2017;
Marchal et al., 1993). Bottom depth has been identified as an additional factor structuring the
vertical distribution of SSLs (Gausset and Turrel, 2001). For example, the thickness and depth
of an SSL on continental shelves tends to increase with an increase in water depth (Torgersen
et al., 1997), similar to patterns observed in deep sea (Berge et al., 2014; Boersch-Supan et al.,
2017). In both deep sea areas and over shelves, the maximum density of SSLs are often
correlated with maximum chlorophyll-*a* concentrations (Berge et al., 2014; Dekshenieks et al.,
2001; Holliday et al., 2010). Dissolved oxygen concentrations (above 1 ml l$^{-1}$) can also predict
the lower boundary SSL density, *e.g.*, in the Eastern Boundary Upwelling Systems (EBUS), the
Peruvian Coastal Upwelling system (Bertrand et al., 2010), and the California Coastal
Upwelling system (Netburn and Koslow, 2015).
In this study, we use acoustic tools (Simmonds and MacLennan, 2005) to examine the fine-
scale vertical structure of SSLs (*i.e.*, their depth in the water column, thickness, and density)
(Bertrand et al., 2013; Perrot et al., 2018). We use fine spatiotemporal resolution of acoustic
data to investigate how the pelagic environment influences SSLs during an upwelling event in
the EBUS off Senegal. Our objective was to model variations in SSLs structure relative to
physiochemical characteristics of water masses and their locations on the shelf.



## 2   Materials and methods

### 2.1   SSLs acoustics sensing and environmental data

We performed a hydroacoustic survey along the "Petite Côte," south of Cap-Vert Peninsula (Senegal) (14.6°–13.5° North, 16.9°–17.6° West). The survey was conducted from the research vessel Antea (IRD, 35m) during the upwelling season from 6–18 March 2013. The Petite Côte is a nursery area for fish and is the main area in which juvenile of numerous species (particularly small pelagic species) concentrate off the Senegal coast (Diankha et al., 2018; Thiaw et al., 2017). Strong upwelling occurs off this coast, which contributes to high primary productivity, thus providing an ideal nursery area for commercially important fish species (Tiedemann and Brehmer, 2017).

We collected hydroacoustic data along three radials (R1, R2, and R3) for 18 nautical miles (nmi) perpendicular to the coast (Fig. 1). We collected acoustic data continuously (day and night) using a Simrad EK60 echosounder (38 kHz), set at 20 log R time-varied gain function (R = range in meters) and used a pulse length of 1.0 ms. Transducers were calibrated following the procedures recommended in Foote et al. (1987). Considering aft draught of the vessel, the acoustic near field and the presence of parasite in the upper part of the water column we have applied an offset of 10 m. Echoes along the three radials were integrated for at a spatial resolution of 0.1 nmi*1m depth. We estimated the SSLs acoustic density by calculating the Nautical Area Scattering Coefficient (NASC or $s_A$), which represents the relative biomass of acoustic targets. We assumed that the composition of the scattering layers and the resulting scattering properties of biological organisms in the SSLs are homogeneous within each layer we identified (*sensu* MacLennan et al., 2002). We analyzed integrated echoes using the in-house tool "Matecho" (Perrot et al., 2018). Matecho is integrative processing software that allows to manually correct echograms (*e.g.*, by correcting bottom depths, removing empty pings, removing echogram interferences, and reducing background noise). After each echogram correction, we extracted the SSLs that were below the mean acoustic volume backscattering strength ($S_v$ in dB) threshold of -75 dB (*i.e.*, values below -75 dB were excluded from the analysis). Cascão et al., (2017) and Saunders et al., (2013) excluded marine pelagic organisms that backscattered at -70 dB, a threshold based on the aggregative behavior of marine pelagic organisms. In our study, the backscattering was due to zooplankton and micronekton which here also include small pelagic fish. The inshore area is known to be rich in fish larva and eggs (Tiedemann and Brehmer, 2017) however, low sample number was collected in the coastal inshore water due to security reason, *i.e.*, research vessel investigate area> 20 m bottom depth).



We collected hydrologic data using a calibrated "Seabird SBE19 plus" conductivity,
temperature, and depth (CTD) probe. The CTD specifications were: for temperature, $\pm 5.10^{-3}$
°C accuracy and $1.10^{-4}$ °C precision; for conductivity, $\pm 5.10^{-4}$ S m$^{-1}$ accuracy and $5.10^{-5}$ S m$^{-1}$
precision; for pressure, $\pm 0.1\%$ of full scale range accuracy and $2.10^{-3}$ % precision of full scale
range precision.  The CTD was equipped with sensors for fluorescence ($\pm 2.10^{-3}$ μg l$^{-1}$ accuracy,
and $\pm 2.10^{-4}$ μg l$^{-1}$ precision) [a measure of chlorophyll-*a* concentration, a proxy for
phytoplankton biomass], and dissolved oxygen (Seabird SBE43, 2% saturation for accuracy
and 0.2% saturation for precision). From 6–8 March 2013, we conducted CTD casts along three
radials at 36 stations. At each station, sensors measured water temperature (°C), depth (m),
fluorescence (μg l$^{-1}$), water density (kg m$^{-3}$), and dissolved oxygen (DO, ml l$^{-1}$). Global High
Resolution Sea Surface Temperature (GHRSST) data were extracted from daily outputs by the
Regional Ocean Modeling System group at NASA's Jet Propulsion Laboratory
(https://ourocean.jpl.nasa.gov/). Daily SST data (GHRSST Level 4 G1SST Global Foundation
Sea Surface Temperature Analysis) were averaged for the three days of surveying using
SeaDAS software version 7.2 (https://seadas.gsfc.nasa.gov/) and interpolated on maps using
the R software (R Core Team, 2016). Cubic spline interpolations of gridded data were used
within the R package Akima (Akima et al., 2016)
2.2    Data analysis
After extracting SSLs with Matecho, we developed an *ad hoc* Matlab program named
"Layer" to calculate layer thickness and another program, "ComparEchoProfil" to fit
echograms to the minimum and maximum depths measured along each CTD vertical station.
The ComparEchoProfil displayed the profile for mean acoustic volume backscattering strength
($S_v$) in dB over a small-scale elementary sampling unit 'ESU' of 0.1 nmi around each CTD
station. The program also allowed us to display acoustic profiles for physiochemical parameters
(temperature, CHL, density, and $O_2$) associated with $S_V$ profiles (Fig. 2). The output included
meta information [station ID, station date, station time, latitude and longitude, diel phase (day,
night), and local shelf depth (bottom depth)], all of which we associated with SSLs descriptors
[SSL thickness, maximum SSL depth, and the nautical mean backscattering coefficient ($s_A$)]
and physiochemical parameters associated with each SSL.
We used R software (R Core Team, 2016) to statistical analyze and map data. We used the
R package 'Cluster' (Maechler et al., 2014) for Hierarchical Cluster Analyzes (HCA) of CTD
data, the R package 'maps'(Brownrigg, 2016) to map stations, the package 'ade4' (Chessel et





al., 2013) to run Principal Component Analysis (PCA), and the package 'oce' (Kelley, 2015)
to display vertical section plots of physiochemical parameters.

We applied HCA to discriminate between water masses of inshore and offshore stations over

the continental shelf based on CTD data collected at 10 m depth. We used PCA (Chessel et al.,
2013) on the same dataset to determine similarities between CTD stations relative to
environmental parameters. Physiochemical parameters were standardized *a-priori* because they
were measured with different metrics.

Longitudinal variability (*i.e.,* inshore - offshore area) of morphometric (thickness, depth)

and acoustic characteristics ($s_A$) of the SSLs are investigated in the discriminated groups
considering bottom depth and diel period. Diel transition periods are removed from analyses to
avoid SSL density changes bias due to diel vertical migrations. Transition periods are defined
using sun altitude, *i.e.*, around sunset and sunrise corresponding to a sun altitude between 18°
and +18° (Lehodey et al., 2015). Morphometric and acoustic characteristics of the SSLs are
also compared between the inshore area *versus* offshore area, and between day and night using
student's t-test whose application conditions have been verified (normal distribution and
variance equality).

Echogram *vs.* profile coupling figures (Fig. 2 resulting from the "ComparEchoProfil" were

analyzed to determine the relation between environmental parameters and SSLs. An ANCOVA
(analysis of covariance) (Wilcox, 2017) was implemented to build a model to predict the SSLs
structure (thickness, depth, and density). The selection of the best model was performed using
stepwise procedures. Stepwise selection was based on the Akaike Information Criteria (AIC) ;
best model choice is based on the minimum value of AIC (Akaike, 1974). The relative
importance of each variable in the total deviance was determined from the "relaimpo" R
package (Tonidandel and LeBreton, 2011). Validity assumptions of the models was then
assessed by checking for normality of distributed errors and homogeneity of residuals
(Appendix A to C). For the ANCOVA, SSL density and $s_A$ was log10 transformed for normality
assumption. For all statistical tests, the significance threshold used was 0.05.
## 3   Results
### 3.1   Characterization of two water masses over the shelf

The HCA differentiated two groups of stations (Fig. 3a): Group 1 (G1) stations (n =18) was

comprised of four stations along radial R1, six stations along radial R2, and eight stations along
radial R3. The G1 stations were located closest to the coast (inshore area, from 13 to 61 m
bottom depth, which encompassed the core of the upwelling (based on data for sea surface



temperature) (Fig. 1). Group 2 (G2) stations (n = 18) comprised seven stations along radial R1,
six stations along radial R2, and five stations along radial R3. These stations were located
furthest from shore (offshore area), from 41 to 205 m bottom depth, which corresponds to the
outer border of the upwelling zone. Considering the bathymetry, we note an overlay of the two
areas discriminated between 41 to 61 m.

PCA also identified the same two distinct water masses that the HCA identified (Fig. 3).

Axis 1 of the PCA eigenvalues explained 72.8% of the inertia, whereas Axis 2 explained 26.8%.
On Axis 1of the PCA plot, temperature was highly correlated to density. On Axis 2,
temperature, and $O_2$ were opposed to CHL. The distribution of these variables are related to the
station groupings: G1 (inshore area) was characterized by a dense and CHL-rich water mass,
whereas G2 (offshore area) was characterized by a warm and slightly oxygenated surface water
mass.

Satellite measurements of SST distributions of the study area indicated the same split of

stations into two groups (Fig. 4). The inshore area was characterized by low SST values (18–
19 °C), indicating a recently upwelled water mass, whereas an older water mass with higher
SST values (20–21 °C) prevailed offshore. SST revealed an advection of water masses from
offshore to inshore responsible for a convergence of the two water masses.

At radial R1, a marked frontal zone appeared isolating two water masses between 20 – 40 m

isobaths (Fig. 5a1) which separate warm surface waters from deep cold waters. At radials R2
and R3, the upwelling appear as a cold-water tongue isolating a warm water band at the coast
(Fig. 5a2, a3). At R3, this cold-water tongue was expanding toward the inshore area as well as
to the offshore area (Fig. 5a3). Surface waters of the inshore area are slightly denser than water
masses in offshore area with approximately 26 kg m$^{-3}$ and 25 kg m$^{-3}$, respectively. For CHL,
elevated concentrations were exclusively observed in the inshore area at radials R1 and R2. CHL
was significantly higher in the inshore area than the offshore area with concentrations of 3.0 –
5.0 mg m$^3$ in the inshore area to 0.3 – 2.0 mg m$^3$ in the offshore area (Fig. 5c). At R3, the
elevated CHL concentrations were observed in both inshore and offshore area close to the
upwelling front. CHL was also higher in the upper part of the water column (0 – 20 m)
decreasing with depth in both area. Higher DO concentrations were observed towards both sides
of the upwelling core. At R1, the upwelling front was at the most coastal part separating the
inshore area from the less oxygenated offshore area with DO concentrations of 5.0 – 7.0 ml l$^{-1}$
and 4.0 – 5.0 ml l$^{-1}$, respectively. At R2 and R3, the core moved towards the offshore, separating
the inshore area (DO concentrations of 4.0 – 5.0 ml l$^{-1}$) slightly more oxygenated than the





offshore area (DO concentrations of $2.0 - 4.0$ ml $l^{-1}$). DO concentration decreased also from the
surface to bottom in both areas.
3.2    Variability in vertical structure of SSLs
3.2.1    Spatial variability according to water mass characteristics
Thickness and depth of the SSLs varied according to bottom depth in the inshore area and
the offshore area. In the inshore area, on the northern radial R1, no SSLs was observed at
coastal stations below 29 m depth (stations 1 and 2) (Fig. 6a). In offshore stations, starting
at 41 m depth, the SSLs were observed in all stations and radials (Fig. 6b), and their thickness
and depth increased with bottom depth. SSL thickness and SSL depth differed significantly
between the inshore area and the offshore area: the SSLs were thicker and deeper in the
offshore area than in the inshore area (Fig. 7) ($p$-value $= 0.001$ for both thickness and depth).
An increase of SSL density ($s_A$) was observed with bottom depth in the inshore area and the
offshore area. The $s_A$ comparison between the inshore area and the offshore area (Fig. 7) was
not significantly different ($p$-value $= 0.833$).
3.2.2    Diel vertical migration
The diel period had a significant effect on SSL thickness ($p$-value $< 0.001$), and SSL
depth ($p$-value $< 0.001$) which were found both higher during the night in the inshore area
and the offshore area (Fig. 7). In the inshore area (13 to 61 m bottom depth), during the day,
SSLs had a mean depth of $19 \pm 11$ m and a mean thickness of $11 \pm 8$ m, while during night-
time SSLs were observed in mean depths of $46 \pm 15$ m ($\Delta_{Night-Day} = 27$ m) having a mean
thickness of $35 \pm 15$ m ($\Delta_{Night-Day} = 24$ m). In the offshore area (41 to 205 m bottom depth),
SSLs were found at a mean depth of $49 \pm 11$ m during daytime with a mean thickness of 38
$\pm 11$ m, while during night-time SSLs depth was $86 \pm 22$ m ($\Delta_{Night-Day} = 37$ m) with a mean
thickness of $75 \pm 22$ m ($\Delta_{Night-Day} = 37$ m).
Mean $s_A$ of SSLs varied between day and night (Fig. 7) but were not significantly different ($p$-
value $= 0.890$). In the inshore area, $s_A$ was $24 \pm 23$ m$^2$ nmi$^{-2}$ during the day and $44 \pm 59$ m$^2$ nmi$^{-}$
$^2$ during the night ($\Delta_{Night-Day} = 20$ m$^2$ nmi$^{-2}$). In the offshore area, mean $s_A$ of $46 \pm 4$ m$^2$ nmi$^2$
during daytime was not significantly different from mean $s_A$ of $25 \pm 7$ m$^2$ nmi$^2$ during night-
time ($\Delta_{Night-Day} = 21$ m$^2$ nmi$^{-2}$).





### 3.3 Effect of environmental parameters on SSLs

#### 3.3.1 Vertical dimension of SSLs related to physicochemical profile

In both areas, SSLs were partially or completely located in area of strong vertical gradients of temperature (thermocline), density (pycnocline), and DO (oxycline) (Fig. 2). When a strong temperature gradient - usually also associated to the vertical position of the oxycline and pycnocline is reported a peak of CHL was often observed and match with volume backscattering strength ($S_v$) peak (Fig. 2a). This observation is well illustrated in CTD stations 12, 13, 16, and 25 (Appendix D). In the inshore area, the peak of CHL concentration was always located above the SSLs (Fig. 2a), whereas in the offshore area, the peak of CHL concentration was either above the SSLs or in the middle of the SSLs (Fig. 2b). The thickest SSLs were observed in the offshore area where temperature, density and oxygen gradient were strong.

#### 3.3.2 Behavior of the SSLs relative to pelagic habitat characteristics

##### 3.3.2.1 In the inshore area (G1)

In the inshore area (G1), the model indicated a strong effect of bottom depth and diel period on both SSLs thickness and depth. For SSL thickness, the model (Table 1) explained 87% of the variance ($R^2 = 0.869$, *p*-value = 0.001). Bottom depth explained 56 % of SSL thickness while the diel period effect accounted for 31%. The model of SSL depth (Table 2) was like those of SSL thickness, *i.e.*, the model included bottom depth and diel period explained 80 % of the variance ($R^2 = 0.805$, *p*-value = 0.001). Bottom depth showed the largest effect on SSLs explaining 51% of SSL depth while diel period effect was estimated at 30%. For SSL acoustic density, *i.e.,* $\log(s_A)$ (Table 3), the model explained 40% of the variance ($R^2 = 0.398$, *p*-value=0.022) indicating a single effect of bottom depth on $\log(s_A)$ (*p*-value=0.020). The bottom depth was the only variable significant in the model and explained 33% of SSL acoustic density Temperature was insignificant in the model.

##### 3.3.2.2 In the offshore area (G2)

For offshore stations, the model showed a significant effect of diel period, temperature, water density and DO on both thickness and depth of SSLs with similar results. Both models, SSL thickness (Table 1) and SSL depth (Table 2) included bottom depth, diel period, temperature, density and DO explaining 85 % of variance ($R^2 = 0.855$, *p*-value = 0.001). Bottom depth and diel period accounts for 28.0% and 28.3%, respectively. Other significant variables are water temperature, density and DO, which support 11%, 10%, and 7%, respectively. For SSL density or $\log(s_A)$ (Table 3), none of predictor variable has a significant effect.





## 4   Discussion
### 4.1   Characterization of water masses along the Petite Côte
The upwelling phenomenon is a key process in the functioning of the coastal ecosystem of
Senegal and Mauritania (Capet et al., 2016; Estrade et al., 2008; Rebert, 1983). By
characterizing the physiochemical parameters of the Petite Côte, we were able to discriminate
two water masses, an inshore area and the offshore area, both of which could also be
distinguished with SST satellite data.
The spatial structure of SSTs helped us understand the upwelling dynamics along the Petite
Côte. This SST pattern, clearly measured at the time of our sea survey, has been reported in
prior studies as well (Capet et al., 2016; Faye et al., 2015; Ndoye et al., 2014). Wind variability,
topography, and density stratification are the main environmental drivers generating upwelling
(Estrade et al., 2008). During the upwelling season (in winter and late spring), northerly trade
winds induced  strong upwelling core south of Dakar(Ndoye et al., 2014; Roy, 1998). The
upwelling core was over the shelf, and SST was lowest on the coastal side of the shelf break,
increasing in both offshore and coastal directions. This displacement of the upwelling core
towards the middle of the shelf strongly influenced the spatial structure of the surface thermal
field. A tongue of cold water over the shelf isolated a coastal band of warm water from the
offshore area, and there was a surface divergence associated with the upwelling source over the
shelf and convergence nearshore. As upwelling got stronger, its core was moving towards the
shelf edge and coastal warming is enhanced by the convergence occurring nearshore (Roy,
1998). The upwelling core was typically found 10 to > 20 km away from the coast generic for
a wide and shallow continental shelf (Estrade et al., 2008). The spatial difference of CHL
concentration between the inshore area and the offshore area was the result of upwelled water
carrying nutrients at the coast limited by water mass fronts. Nutrient-rich water, supplied to the
sunlit surface layer by wind-driven upwelling stimulates the growth of phytoplankton that
ultimately fuel diverse and productive marine ecosystems (Jacox et al., 2018). There was a link
between the accumulation of biological material and the location of the coastal band of warm
water. This coastal band between coast and the upwelling core have been regarded to function
as retention area in which nutrient particles are trapped (Demarcq and Faure, 2000; Roy, 1998).
The nutrient utilization was optimized by retentive physical mechanisms in the coastal area,
which enhance microbial remineralization of particulate organic matter and zooplankton
excretion, and then regenerated production through ammonium consumption (Auger et al.,
2016). This caused an increase in primary production and results in a surplus of phytoplankton




biomass in inshore area. Low DO concentration observed in the upwelling core separating more
oxygenated water masses have been reported in previous studies (Capet et al., 2016; Teisson,
1983) over the Petite côte. Once a water mass becomes isolated from the atmosphere, its oxygen
content starts to decrease due to the biological remineralisation of the dissolved organic matter
(Emerson et al., 2008; Machu et al., 2019). These low-oxygen bottom waters are transported to
the inner shelf during upwelling favourable wind events. Moreover, temporal stability of the
upwelling core was also noticeable over periods of several days to weeks; and export from the
shelf to the open ocean is retarded (Capet et al., 2016). Thus, in such favorable condition of
continuous food supply, photosynthesis may have fostered an enrichment of DO in the inshore.
This was in line with high CHL levels observed towards both side of upwelling core,
particularly in the inshore area.
4.2    Spatial variation of the SSLs off the Petite Côte of Senegal
We measured a longitudinal gradient in the thickness of the SSLs over the continental shelf.
The SSLs was concentrated in a narrow band in the inshore area, whereas the SSLs was wider
in the offshore zone. The absence or weakness of SSLs in the inshore area (in contrast to the
more stratified water column in the offshore area) may have been due to turbulence in the water
column (Sengupta et al., 2017), coupled with well-mixed surface water. In the inshore area,
turbulence and the probable low residence time of marine pelagic organisms advected from
outside this area, both inhibited the SSLs formation. Indeed, in such upwelling system, in
addition to the retention mechanism that has been recognized by several authors (Arístegui et
al., 2009; Capet et al., 2016; Mbaye et al., 2015; Roy, 1998), there is also an offshore Ekman
transport mechanism (Arístegui et al., 2009; Estrade et al., 2008) that contribute to cross-shore
exchanges. Otherwise, many authors have stressed that SSLs need stable hydrological
conditions to form (Aoki and Inagaki, 1992; Baussant et al., 1992; Marchal et al., 1993; Urmy
and Horne, 2016). For example, in Monterey Bay (California), Urmy and Horne (2016)
observed a decline in backscatter intensity in the upper water column immediately following
an upwelling event. Aoki and Inagaki (1992) found that when the vertical temperature gradient
became weaker in frontal region between warm and cold waters off the northeast coast of Japan,
SSLs formed deeper and became thicker. Therefore, we believe that the increase of SSL
thickness with depth from inshore to offshore off Senegal is caused by upwelled waters that
disrupt the vertical stability of the water column.



### 4.3 Diel temporal variation of SSLs

In our study area, the diel period consistently exhibited pronounced effects on SSLs thickness and depth. Deeper night SSLs have a greater thickness than daytime SSLs. The diel difference of thickness and depth is due to the well-known DVM performed by many marine species. DVM is a behavioural mechanism usually characterized by an ascent vertical displacement during night-time for feeding and a descent to avoid predation by visual predators) during daytime (Bianchi et al., 2013; Haney, 1988; Lehodey et al., 2015). Some plankton and micronekton organisms have been reported to exhibit reverse DVM, *i.e.,* ascending in the morning and descending in the evening or early night, which is the opposite pattern generally observed with migrating animals (Cushing, 1951; Ohman et al., 1983). Previous study reported that DVM of plankton may increase coastal retention in the inshore area (Brochier et al., 2018; Rojas and Landaeta, 2014). Diel variation was also observed for SSL acoustic density which showed opposite patterns in the two area, *i.e.,* higher during night than day in the inshore area and higher during days than night in the offshore area. DVM of marine pelagic organism may not be the only factors causing diel backscatter variations. The acoustic target strength can be strongly dependent on the aspect at which a target is insonified. Target strengths of zooplankton and micronekton can vary by several orders of magnitude between extreme tilt angles, *i.e.,* horizontal *vs*. head up or head down (Benoit-Bird and Au, 2004; Yasuma et al., 2003). Target strength is not independent of depth, as migrations through the hydrostatic depth gradient can alter, *e.g*., swimbladder volume (Fässler et al., 2009). This can bias target strengths, in particular near the resonance frequency, leading to artificial increases of backscatter at particular depth (Davison et al., 2015; Godø et al., 2009; Kloser et al., 2002). However, in the inshore area, the observed DVM type I can be bias, because more CTD station were achieved during the day than night. Otherwise, plankton such as fish larvae are able to perform DVM type II by ascending in the upper 10 m of the water column at night, *i.e*., in the echosounder offset.

### 4.4 Effect of environmental parameters on SSLs

#### 4.4.1 SSLs related to physicochemical parameters in the vertical dimension

Previous studies have shown that hydrologic structures of water column influences SSLs vertical structure (Balino and Aksnes, 1993; Berge et al., 2014; Gausset and Turrel, 2001). Our results showed that vertical distribution of SSLs was linked to strong vertical gradients of temperature, DO and water density (Fig. 2). SSLs were sometimes localised on thermocline, pycnocline and oxycline depth. The depth of SSLs has been described to be related to thermocline (Marchal et al., 1993; Yoon et al., 2007). In more stratified areas, SSLs vertical



distribution was limited by a strong thermocline and when thermocline was not well marked
(low gradient), SSLs occupied the entire water column (Lee et al., 2013). SSLs vertical
distribution is also related to DO (Bertrand et al., 2010; Netburn and Koslow, 2015). DO has
been described as a good predictor of SSLs vertical distribution (Bianchi et al., 2013). Indeed
in case of low DO levels the metabolism of marine organism is often directly affected (Brennan
et al., 2016; Claireaux and Lagardère, 1999; Pörtner, 2010). In our study, DO appeared to have
a limited influence on SSLs vertical position, no doubt due to high DO value in both area
(lowest value at 151 m bottom depth above the general recognition of hypoxia, *i.e.*, DO $< 1.43$
ml l$^{-1}$ (Diaz and Rosenberg, 2008; Keller et al., 2015). Vertical position of SSLs compared to
the CHL concentration peak can be explained by trophic relationships between phytoplankton,
zooplankton and micronekton. It is understood that zooplanktivorous micronekton migrate
upward in the water column to forage on mesozooplankton while the mesozooplankton at the
same time are migrating toward the surface to graze upon the phytoplankton. This trophic
relationship may explain the link in vertical position of the SSLs with the phytoplankton peak
reported in this study.
4.4.2   Behavior of SSLs relative to pelagic habitat characteristics
In the inshore area, where SSLs were sparsely distributed (or sometimes non-existent)
bottom depth and diel period were the main environmental parameters influencing the vertical
distribution (thickness and depth) of the SSLs. Bottom depth has been shown to regulate the
vertical distribution of SSLs in the water column (Donaldson, 1967; Gausset and Turrel, 2001;
Torgersen et al., 1997). In our study, all  stations indicated a single SSL, while in deep water
more thick and deep SSLs often partitioned into multiple layers are observed (Ariza et al., 2016;
Balino and Aksnes, 1993; Cascão et al., 2017; Gausset and Turrel, 2001). Diel period is the
second most important parameter acting on SSL thickness and depth through the DVM
phenomenon. In well mixed water masses, temperature, density and oxygen had no effect on
the SSLs. The non-significant effect of temperature, oxygen, and water density on the SSLs in
the inshore area is explained by the presence of less marked and superficial clines because of
the newly upwelled water. As stated above, SSLs need probably stable condition to occur.
In the offshore area, where vertical gradients were marked, the main parameter structuring
SSL thickness and depth was bottom depth and diel period, but also water temperature, density
and DO. SSLs vertical distribution is known to be a function primarily of temperature (Bertrand
et al., 2010; Hazen and Johnston, 2010; Netburn and Koslow, 2015). Overnight, depth of SSLs
is strongly correlated to the depth of thermal and density gradients (Boersch-Supan et al., 2017;





Cascão et al., 2017; Marchal et al., 1993). In the offshore area, the results suggested that DO
also influenced SSL depth and SSL thickness. In well oxygenated continental shelf waters, DO
influenced SSLs but did not limit their vertical distribution. Some previous work led in French
Polynesia (Bertrand et al., 2000), and in the southern California current ecosystem (Netburn
and Koslow, 2015) showed the Oxygen Minimum Zone (OMZ) acting like a barrier of SSLs
vertical distribution. Bianch et al., (2013) suggested that distribution of open-ocean OMZ may
modulate the depth of migration at the large scale, so that SSLs migrate to shallower waters in
low-oxygen regions, and to deeper waters in well-oxygenated waters.

For both areas, CHL concentration was the only predictor that was not included in any of

the final models. However, coupling echogram *vs*. profile (Fig. 2), we can argue that a relation
between CHL and SSLs exist even if it was not significant in the models, because CHL SSL
biomass peaks match, *i.e.*, always located above or in the middle of the SSLs. Moreover, simple
linear model between CHL and SSLs structure (depth and thickness) was significant in the
inshore area, suggesting that CHL effect on full models was masked by autocorrelation between
predictive variables. Furthermore, the vertical distribution of SSLs can be influenced by the
mixed layer depth (MLD). The MLD is one of the primary factors affecting the vertical
distribution of zooplankton. Lee et al. (2018) have shown that the weighted mean depths of
SSLs exhibit a strong linear relationship with the MLD, meaning that the MLD could be a
significant environmental factor controlling the habitat depth of marine pelagic organisms.
Recent study (Stranne et al., 2018) have also shown that the MLD can be tracked acoustically
at high horizontal and vertical resolutions. The method was shown to be highly accurate when
the MLD is well defined and biological scattering did not dominate the acoustic returns.
However, in our study area with biological scattering dominating the acoustic returns and due
to upwelling event, acoustic methods was not appropriate to determine MLD.

.

## 5   Conclusion

Using our echogram *vs* profile coupling approach, we were able to examine fine-scale

processes affecting SSLs distribution. SSLs were influenced by turbulence level in the
upwelling, which lead to an offshore advection of SSLs organisms. SSLs distribution were
mainly structured by the bottom depth, the diel period and the level of vertical stratification in
water column. SSL acoustic density variation suggested different DVM pattern between the
two areas, *i.e.*, normal DVM in the inshore area and reverse DVM in the offshore area. Such
observation should be considered in modelling exercise to better understand DVM implication



in ecosystem functioning. Further investigations should integrate small-scale turbulence
measurements to better describe the fine scale spatiotemporal variability of SSLs and their
relationship to the pelagic environment.

## 6   Software and Code availability

"Matecho" is an Open-Source Tool available at: https://svn.mpl.ird.fr/echopen/MATECHO/
(login: userecho, password: echopen). Matlab code such as "ComparEchoProfil" and "Layer"
can be shared.

## 7   Sample availability

The public cannot access our data because they belong to the partners who funded the
oceanographic cruise.

## 8   Appendices

Appendix A: Diagnostic diagrams of ANCOVA models between sound scattering layers
(SSLs) depth and environmental parameters (temperature, density, dissolved oxygen,
chlorophyll-a, diel period and bottom depth).

466       A.1. Inshore area (G1)



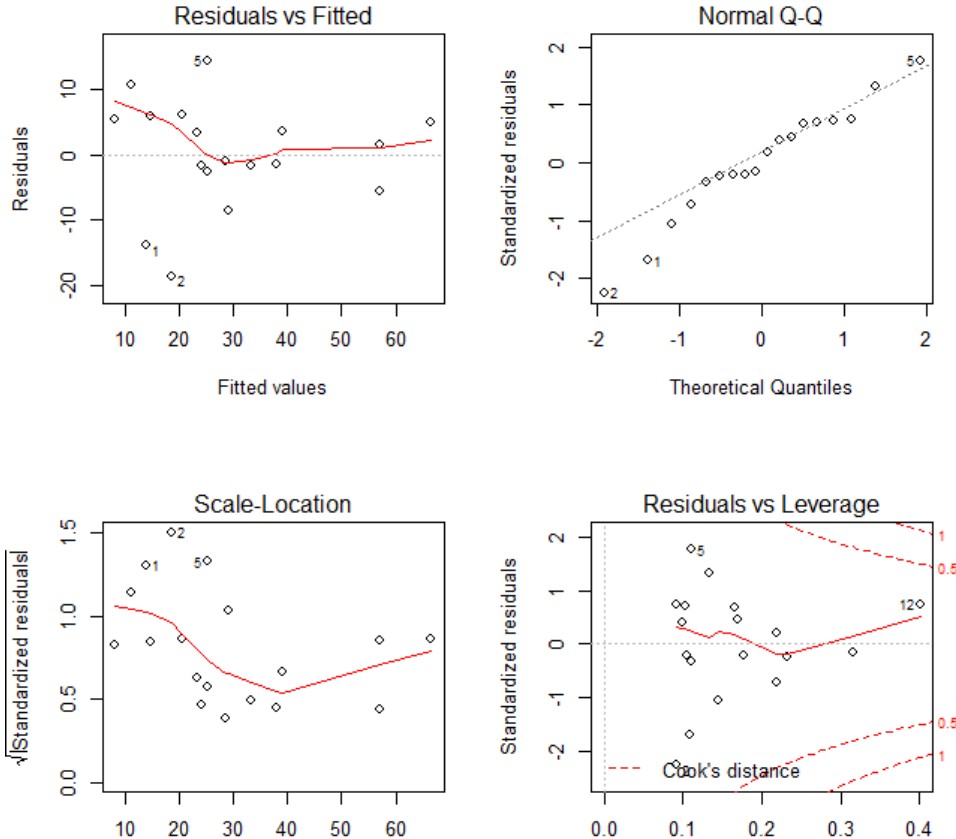






469         A.2. Offshore area (G2)

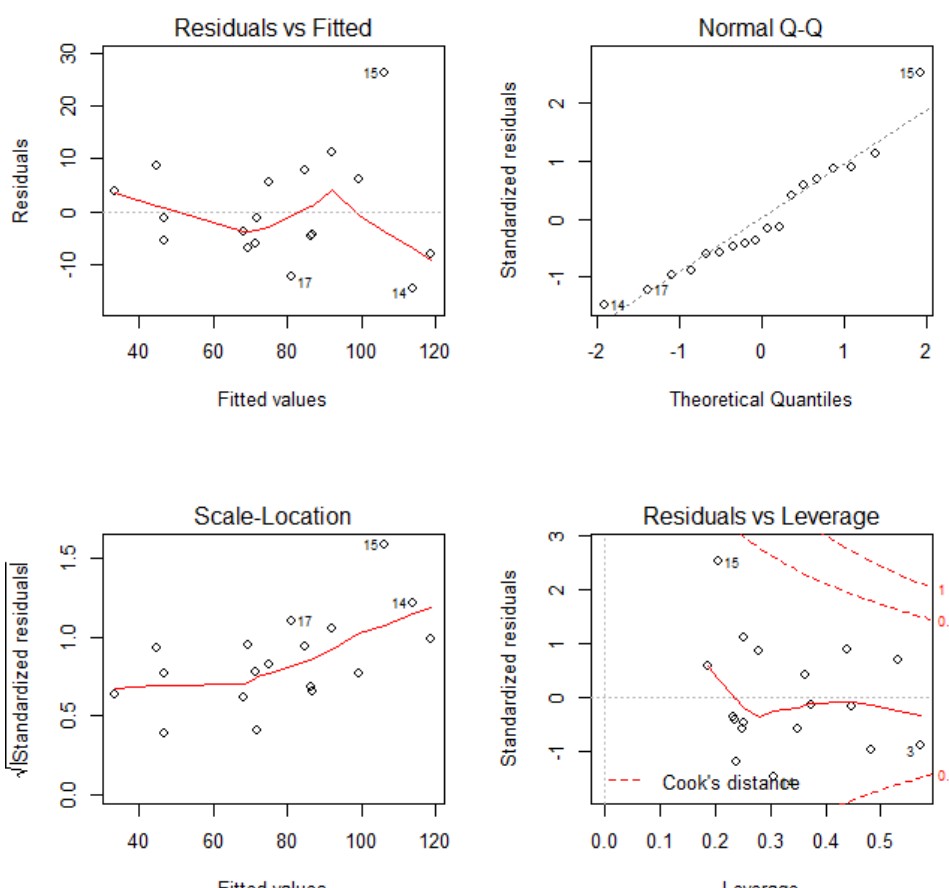





Appendix B: Diagnostic diagrams of ANCOVA models between sound scattering layers
(SSLs) thickness and environmental parameters (temperature, density, dissolved oxygen,
chlorophyll-a and bottom depth.

B.1. Inshore area (G1)


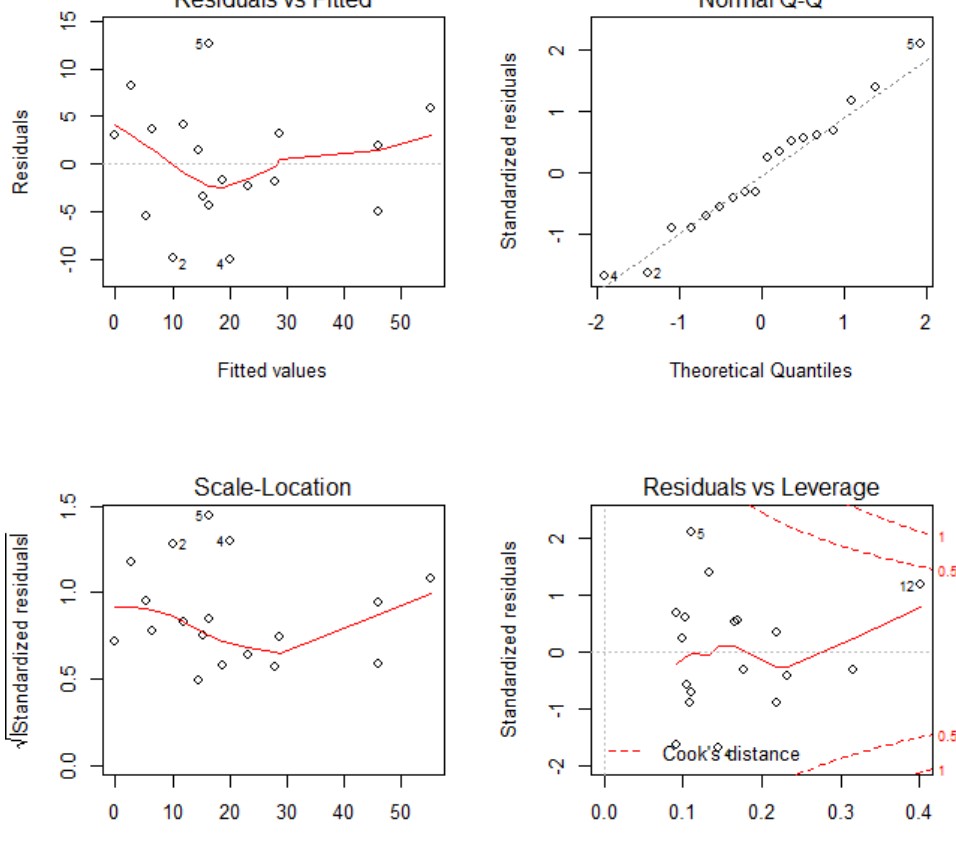



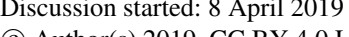



B.2. Offshore area (G2)


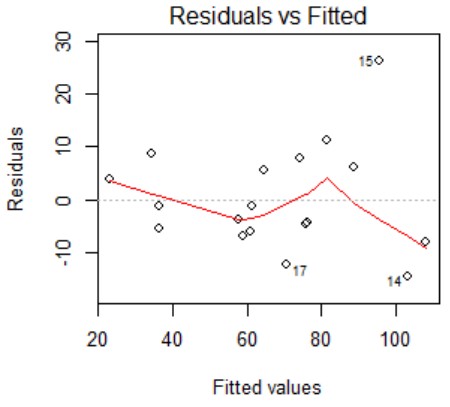
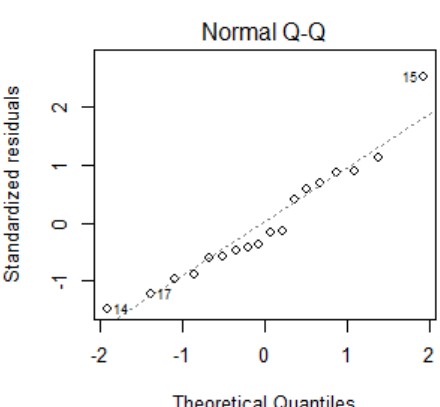

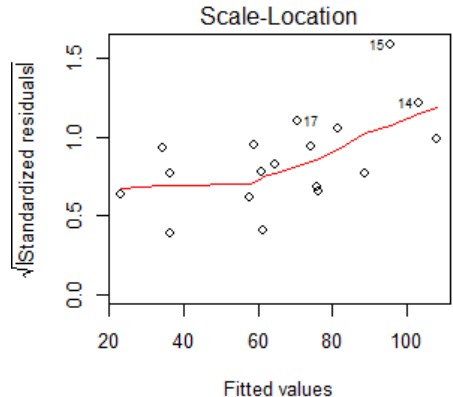
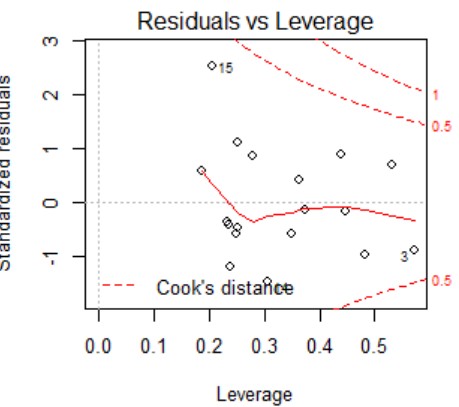





Appendix C: Diagnostic diagrams of ANCOVA models between sound scattering layers
(SSLs) density and environmental parameters (temperature, density, dissolved oxygen,
chlorophyll-a and bottom depth).

C.1. Inshore area (G1)



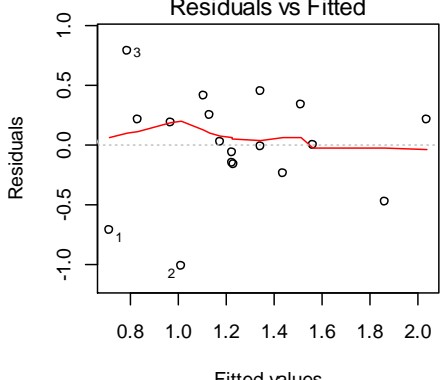
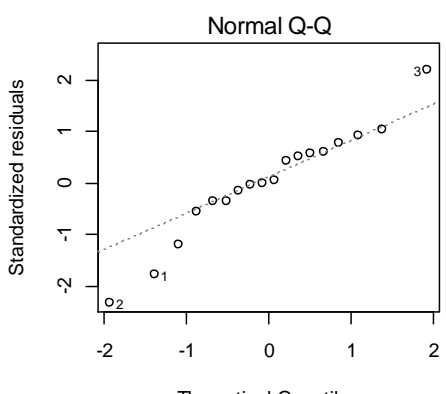
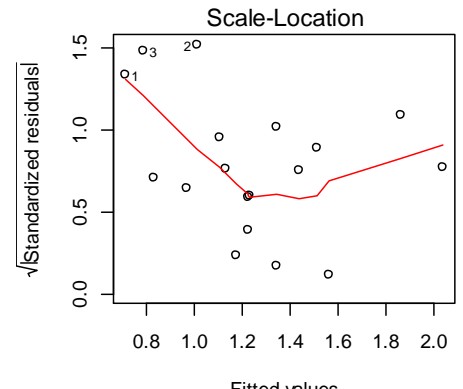
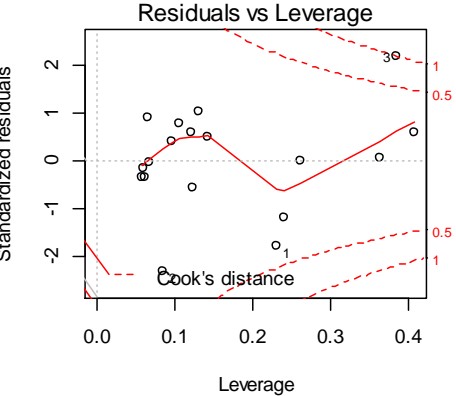






C.2 Offshore area (G2)




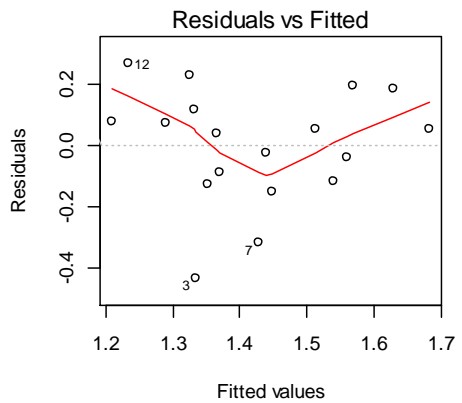

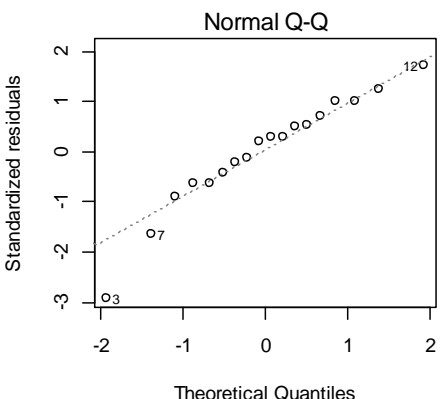

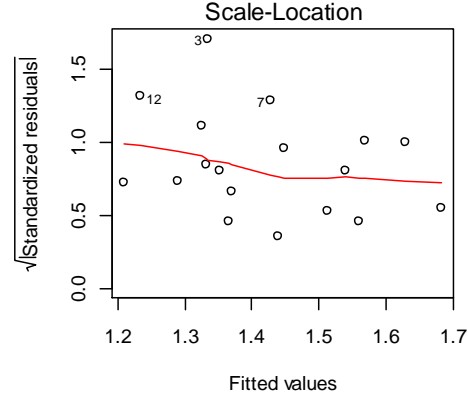

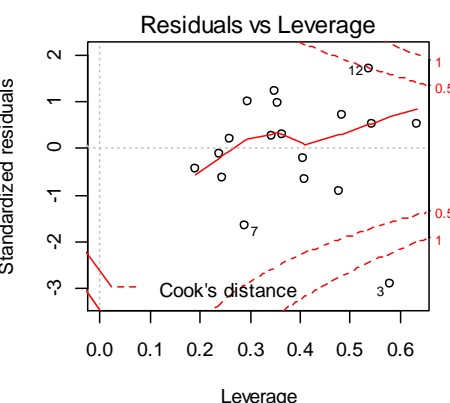




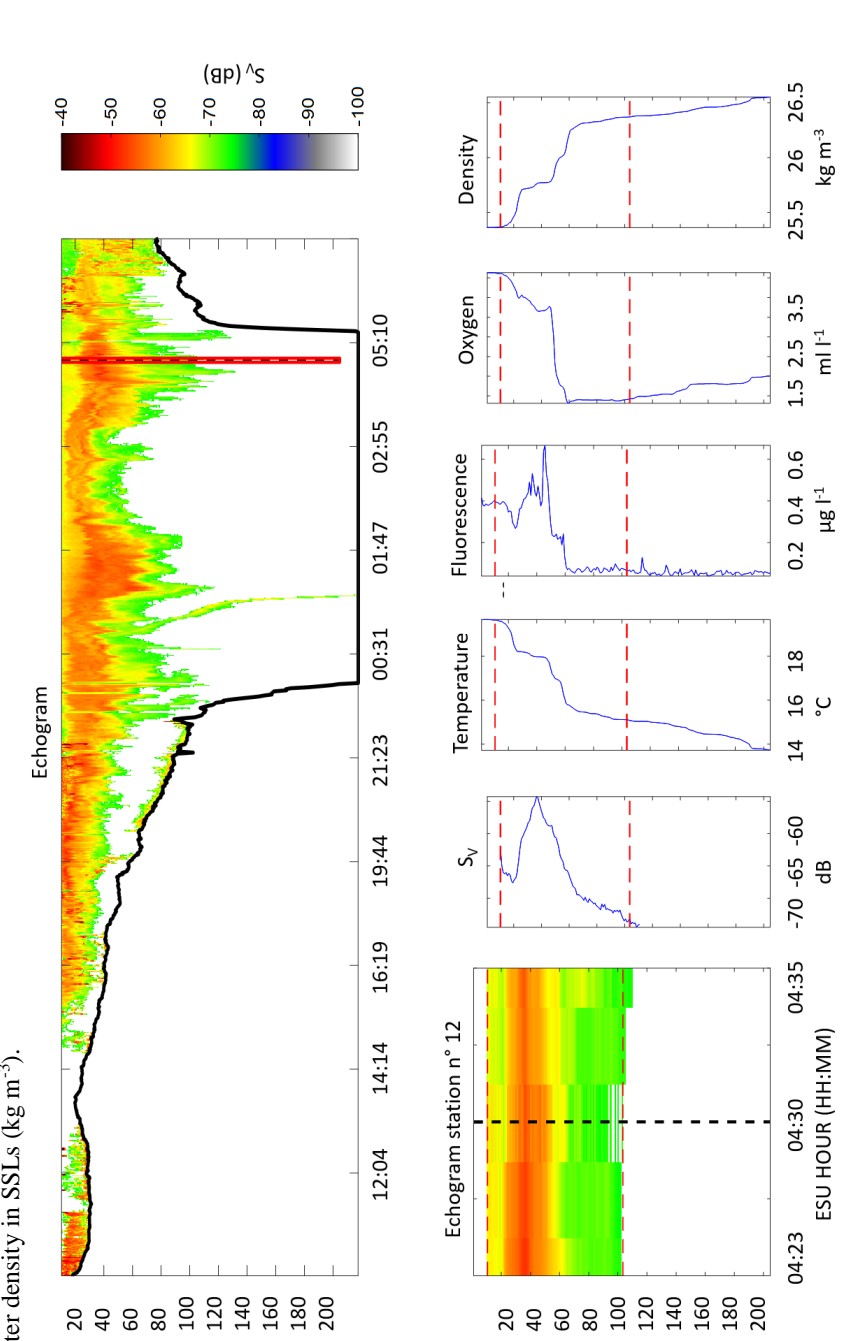

Appendix D: Vertical profile from CTD stations associated to acoustic volume backscattering strength (Sv) integrated per elementary sampling unit (ESU) of 0.1 nmi for 4 station: station 12, 13, 16 and 25. The peak of $S_V$ match the CHL and are related to strong gradient of water temperature, itself related to water density and dissolved oxygen. From the top left to bottom right (i) vertical profile $S_V$ (dB) in the sound scattering layers (SSLs) ; (ii) Profile of mean temperature in SSLs (°C) ; (iii) profile of CHL in SSLs (µg l$^{-1}$) ; (iv) profile of oxygen in SSLs (µmol kg$^{-1}$); (v) and profile of water density in SSLs (kg m$^{-3}$).



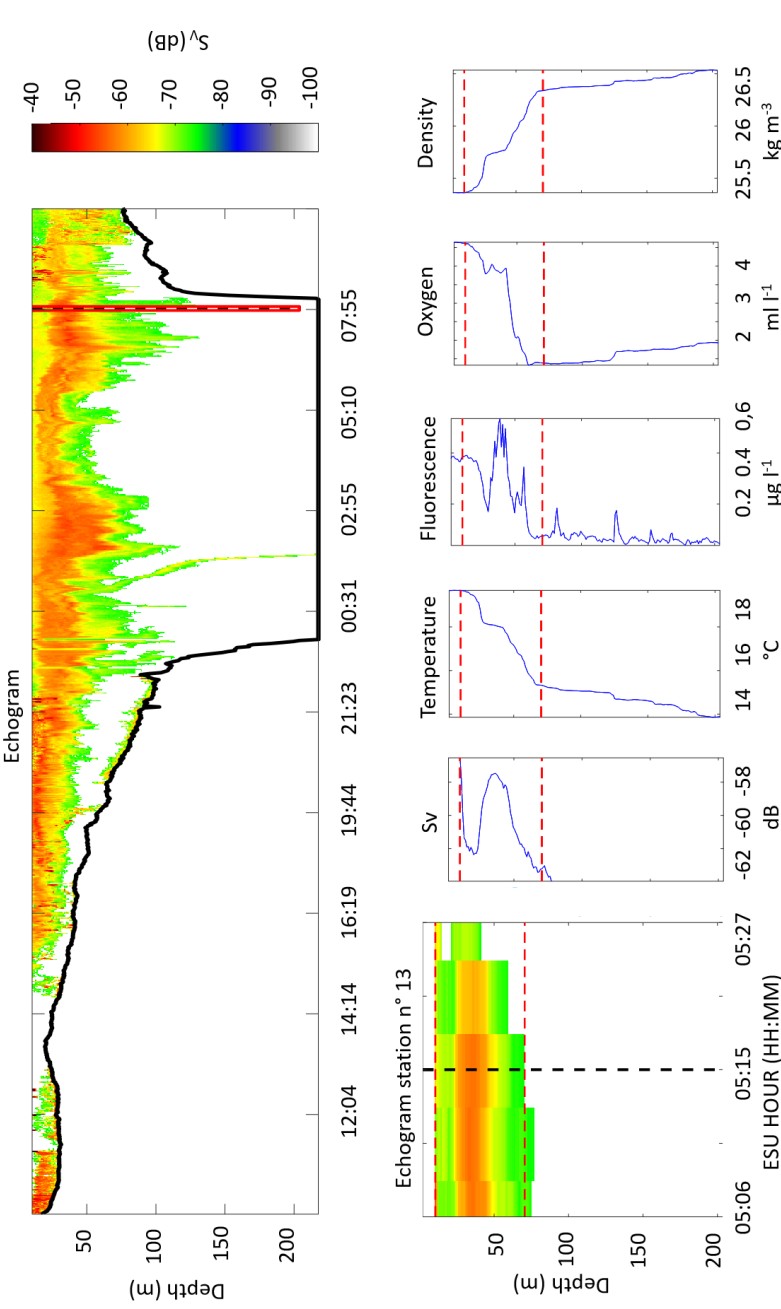



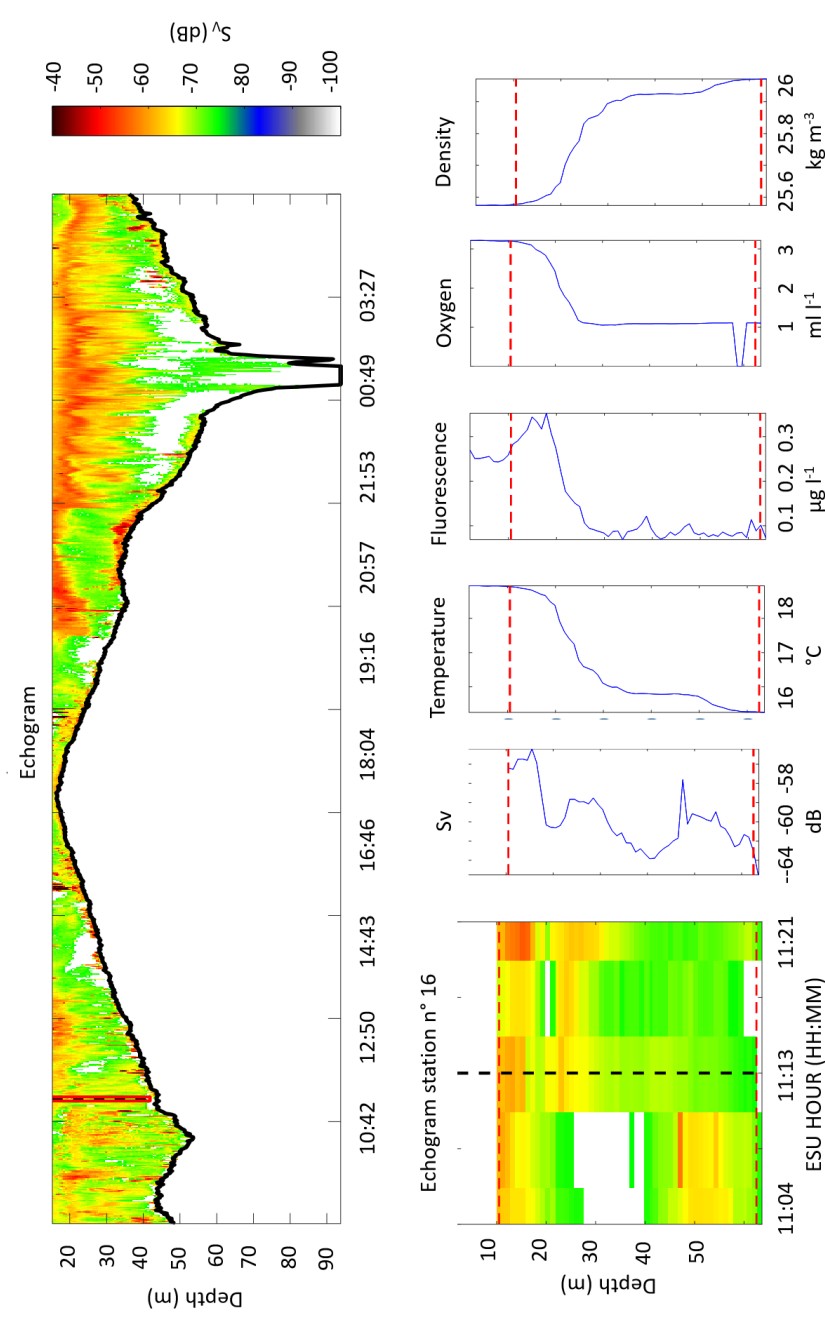



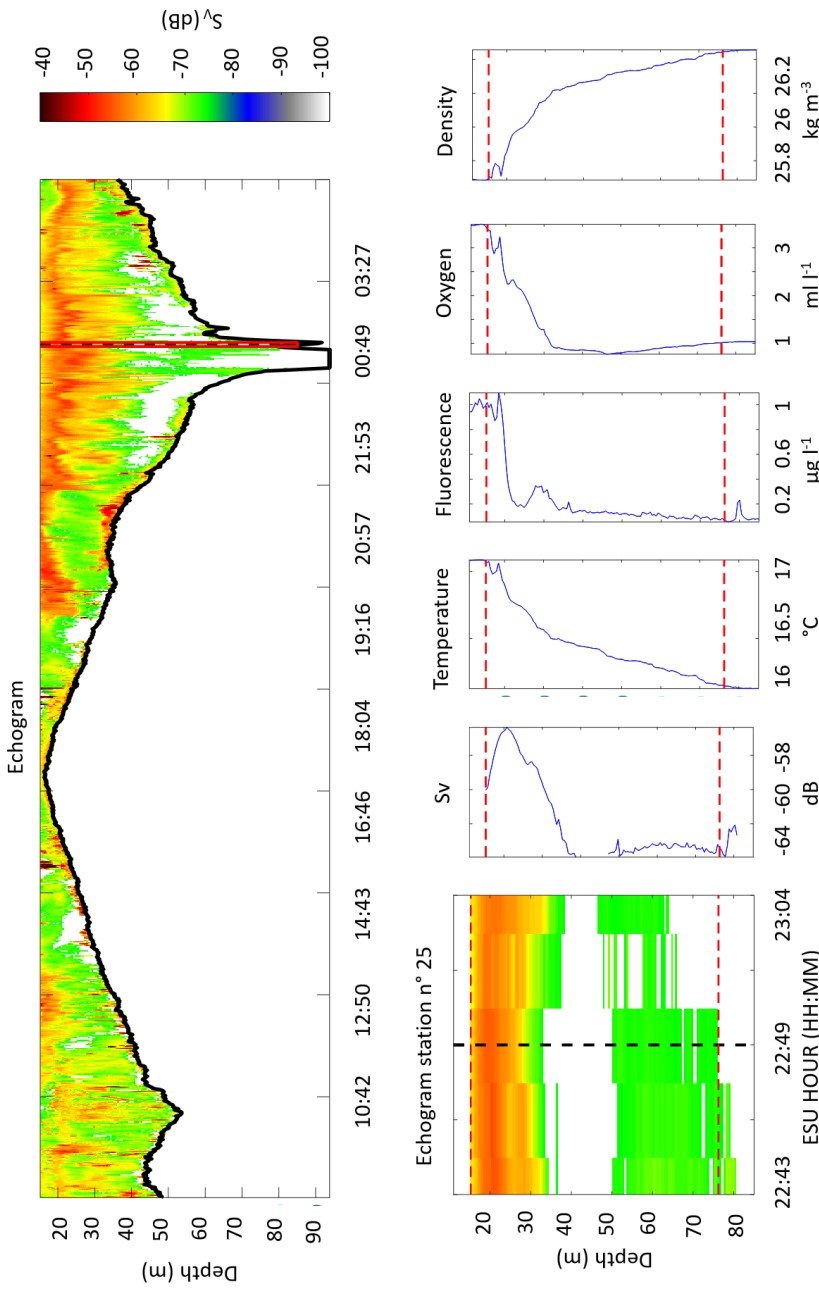



## 9 Author contribution

Ndague DIOGOUL had set the methodology, make the analysis and redacted the paper. Patrice BREHMER defined the methodology, supervised the work and took charge of the acquisition of the financial support for the project leading to this publication. Maik TIEDEMANN helped on data processing and analyze. Yannick PERROT developed "Matecho" software tool and Matlab code. Abou THIAM, Salaheddine EL AYOUBI and Abdoulaye SARRÉ contribute to the redaction. Anne MOUGET, Chloé MIGAYROU and Oumar SADIO helped on statistical analysis.

## 10 Acknowledgments

Results of this paper were discussed during international conferences (ICAWA) in Dakar (2016) and then in Mindelo (2017). We thank participants for helpful comments made during these conferences. We are thankful to the AWA project (Ecosystem Approach to Management of Fisheries and Marine Environment in West African Waters) funded by IRD and the BMBF (grant 01DG12073E), and the PREFACE project (Enhancing Prediction of Tropical Atlantic Climate and its Impacts) funded by the European Commission's Seventh Framework Programme under Grant Agreement number 603521, and all IRD - ISRA/CRODT - Genavir staff helping us at sea during the survey (doi: 10.17600/13110030). We thank Gildas Roudaut, Fabrice Roubaud and the US Imago (IRD) for data collection onboard FRV Antea, the crew of Antea (Gnavir), Dominique Dagorne (IRD) to release satellite products, as well as the personal of ISRA/CRODT (Senegal), IRD DR-Ouest (France) and INRH (Morocco) for their administrative help during Ndague Diogoul PhD stays in Morocco financed by OWSD (Organization for Women in Sciences for the Developing World).

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

## 12  Tables


Table 1: Result of of ANCOVA models between thickness of sound scattering layers (SSLs)
and environmental parameters (temperature, density, dissolved oxygen, chlorophyll-*a,* diel
period and bottom depth) in the inshore area (G1) and the offshore area (G2). [G1: Multiple R-
squared: 0.869, Adjusted R-squared: 0.8515, *p*-value < 0.001]; and [G2: Multiple R-squared:
0.8557, Adjusted R-squared: 0.7956, *p*-value < 0.001]. Significant *p*-value in bold.


| Variable | Significance | | Explained deviance (%) | | Total explained variance (%) | |
|---|---|---|---|---|---|---|
| | Inshore (G1) | Offshore (G2) | Inshore (G1) | Offshore (G2) | Inshore (G1) | Offshore (G2) |
| Bottom depth | **0.001** | **0.005** | 55.86 | 28.05 | 86.9 | 85.57 |
| Diel period (Night) | **0.007** | **0.008** | 31.02 | 28.33 | | |
| Temperature | | **0.007** | | 11.29 | | |
| Density | | **0.008** | | 10.35 | | |
| Oxygen | | **0.007** | | 7.53 | | |



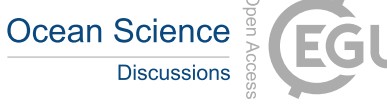

Table 2: Result of ANCOVA models between depth of sound scattering layers (SSLs) and
environmental parameters (temperature, density, dissolved oxygen, chlorophyll-*a,* diel period
and bottom depth) in the inshore area (G1) and the offshore area (G2). [G1: Multiple R-squared:
0.8056, Adjusted R-squared: 0.7797, *p*-value: 0.001]; and [G2: Multiple R-squared: 0.8557,
Adjusted R-squared: 0.7956, *p*-value: 0,000]. Significant *p*-value in bold.

| Variable | Significance | | Explained deviance (%) | | Total explained variance (%) | |
|---|---|---|---|---|---|---|
| | Inshore (G1) | Offshore (G2) | Inshore (G1) | Offshore (G2) | Inshore (G1) | Offshore (G2) |
| Bottom depth | **0.001** | **0.005** | 55.86 | 28.05 | 80.56 | 85.57 |
| Diel period (Night) | **0.021** | **0.008** | 31.02 | 28.33 | | |
| Temperature | | **0.007** | | 11.29 | | |
| Density | | **0.008** | | 10.35 | | |
| Oxygen | | **0.007** | | 7.53 | | |





Table 3: Result of ANCOVA models between sound scattering layers (SSLs) density
$(\log(s_A))$ and environmental parameters (temperature, density, dissolved oxygen, chlorophyll-
*a,* diel period and bottom depth) in the inshore area (G1) and the offshore area (G2). [G1:
Multiple R-squared: 0.398, Adjusted R-squared: 0.3178, *p*-value: 0.022]; and [G2: Multiple R-
squared:  0.3448, Adjusted R-squared: -0.01258, *p*-value: 0.490]. Significant *p*-value in bold.

| Variable | Significance | | Explained deviance (%) | | Total explained variance (%) | |
|---|---|---|---|---|---|---|
| | Inshore (G1) | Offshore (G2) | Inshore (G1) | Offshore (G2) | Inshore (G1) | Offshore (G2) |
| Bottom depth | **0.008** | 0.357 | 33.06 | 7.56 | 39.8 | 34.48 |
| Temperature | 0.119 | 0.273 | 6.73 | 5.17 | | |
| Diel period (Night) | | 0.007 | 0.546 | 7.22 | | |
| Density | | 0.008 | 0.250 | 5.56 | | |
| Oxygen | | 0.007 | 0.166 | 5.19 | | |


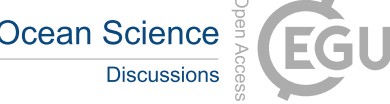

## 13 Figures

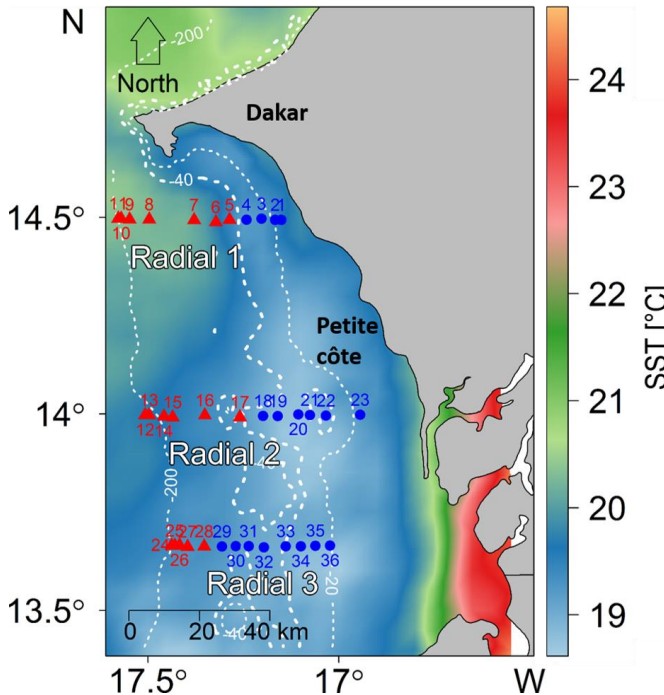

Fig. 1: Location of the survey area off the South Senegal (West African) coast. The hydroacoustic survey was conducted aboard the FRV Antea (IRD) from Dakar (Cap Vert peninsula in the north) to the northern border of Gambia (horizontal black line). CTD-probes collected data at stations along three radials perpendicular to the coast (R1 to R3). Sea surface temperature (SST in degree Celsius) were averaged over the three days of CTD sampling from the 6–8 March 2013. Stations of Group 1 (blue circles) occurred in the inshore zone, whereas stations of Group 2 (red triangles) were situated more offshore



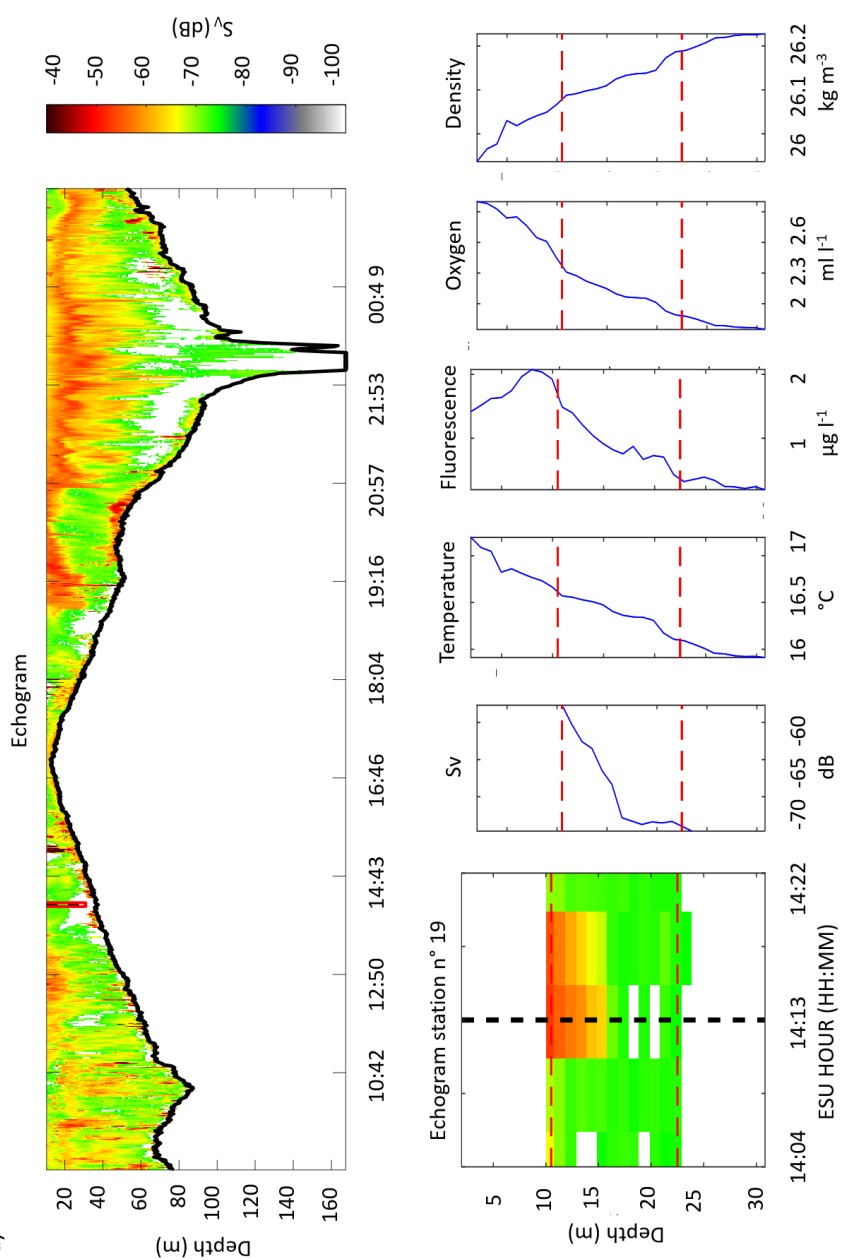

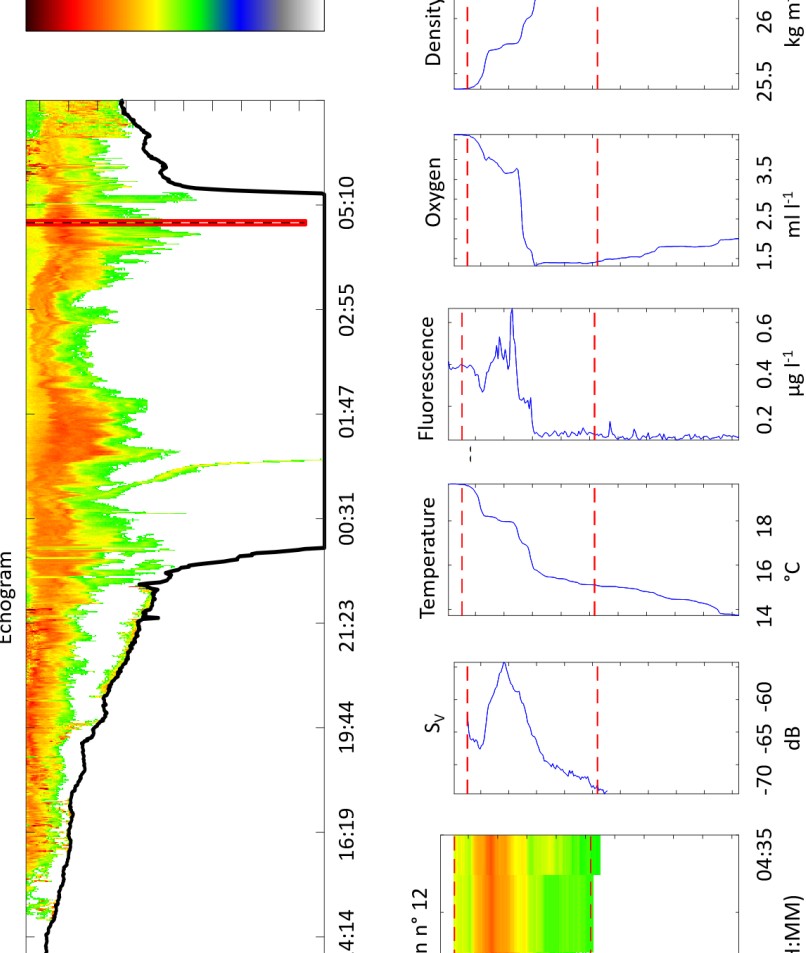




Fig. 2: Echograms and associated vertical acoustic profiles and physiochemical parameters (CTD data) for two example of station: (a) station 19 in the "inshore area" and (b) station 12 in the "offshore area". For both (a) and (b), top panels are echogram data collected along the radial (nmi), whereas the bottom panels depict acoustic and environmental data (depicted by the red vertical line in top panels). Environmental data for the sound scattering layer (SSL) collected at the stations at the time depicted by dotted vertical lines. Data represent mean conditions for the station collected within an area of 0.1 nmi area around the station: acoustic volume backscattering strength ($S_v$) SSL, temperature profile SSL, CHL profile SSL, oxygen profile SSL, and density profile SSL.

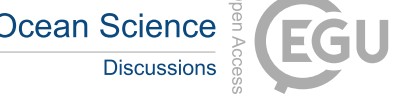

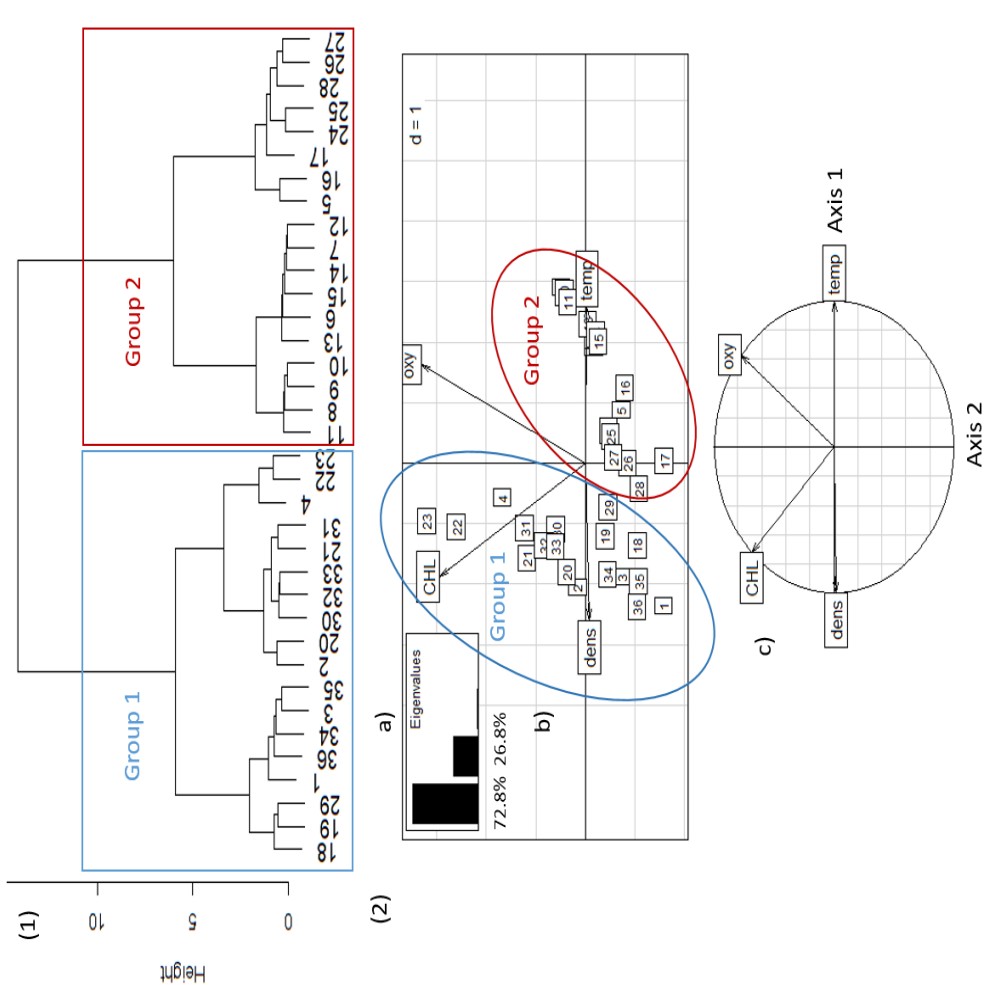



Fig. 3: Discrimination of 36 CTD stations off the Senegal coast: (1) Two groups of stations were discriminated based on data for temperature (temp), chlorophyll-*a* (CHL), dissolved oxygen (oxy), and density (dens). (2) Principal Components Analysis of environmental parameters for all 36 stations. (a) Eigenvalue diagram; (b) Factor plane; (c) Correlation circle. Group 1 are stations located in the inshore area (n = 18), Group 2 are stations located in the offshore area (n = 18).



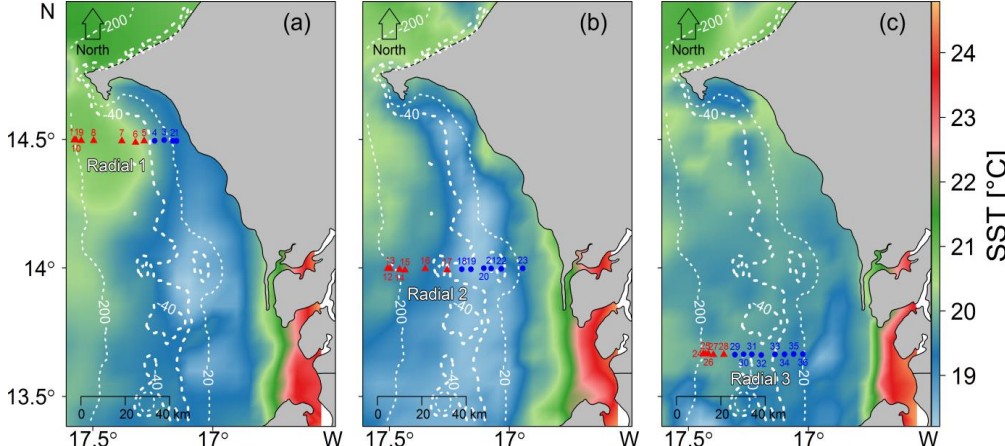

Fig. 4: Positions of vertical CTD stations sampled with a CTD instrument. Diagrams depict temperature, density, fluorescence, and dissolved oxygen relative to daily maps of Sea Surface Temperature (SST) off the Petite Côte (Senegal, West Africa) during the 2014 hydroacoustic survey. (a) Stations along Radial 1 (6 May), (b) stations along Radial 2 (7 May), and (c) stations along Radial 3 (8 May). The blue points are locations for stations of Group 1 (inshore area); red points are locations for stations of Group 2 (offshore area), discriminated according to CTD values measured at 0–10 m depth.





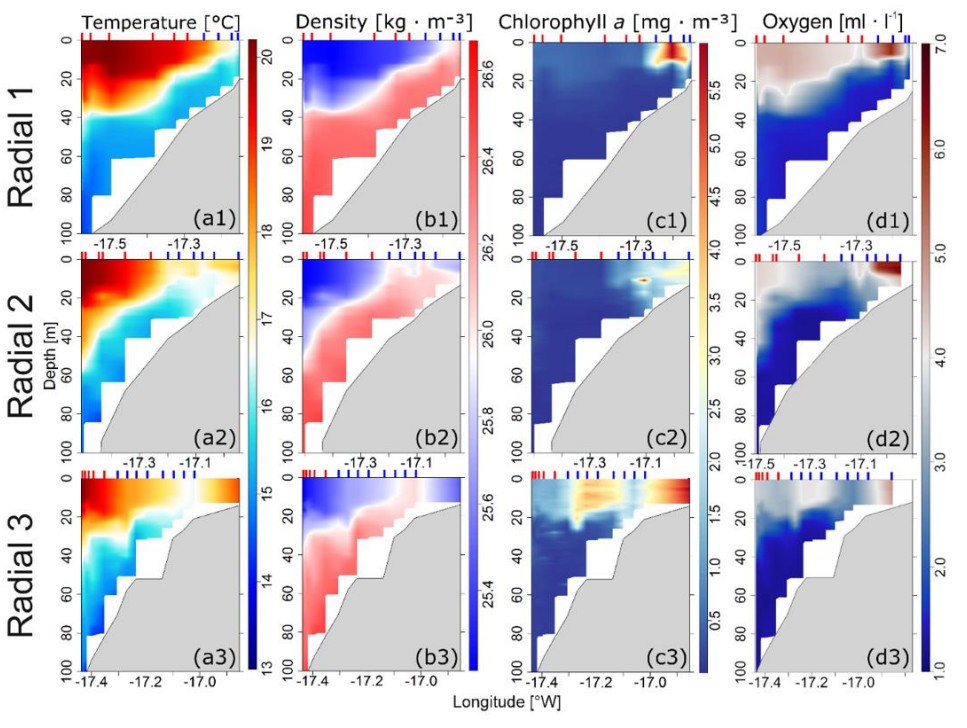

Fig. 5: Mean vertical profile of (a) temperature, (b) density, (c) chlorophyll-a
concentration, and (d) dissolved oxygen in the three radials (R1, R2, R3; see Fig. 1)
with positions of vertical probe stations CTD in the inshore area (vertical line in blue
(G1)) and the offshore area (vertical line in red (G2)).




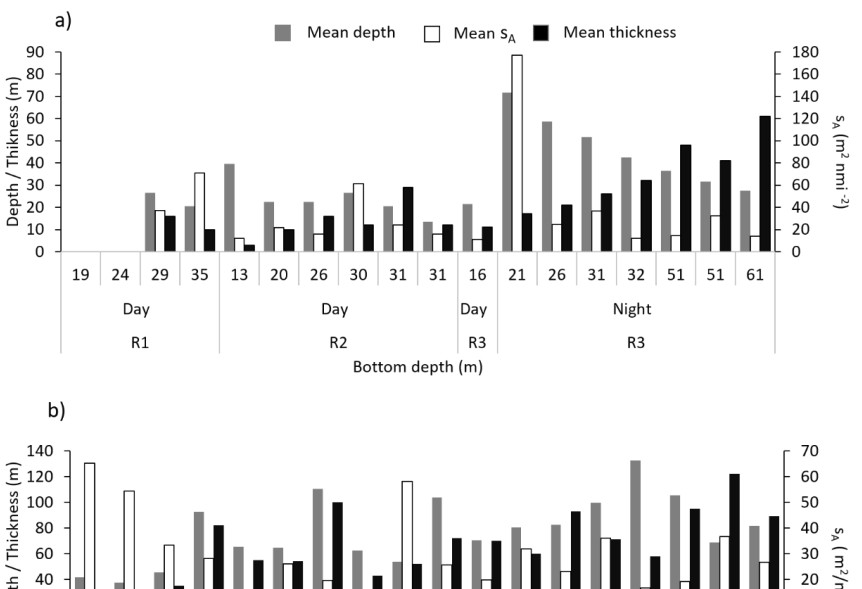

Fig. 6: Variations in mean depth of sound scattering layers (SSLs) (in grey),
thickness of SSL (in black), and SSL mean Nautical Area Scattering Coefficient
(NASC) (in white), in (a) inshore area and (b) offshore area, and according to the
bottom depth along radials R1, R2, and R3 (Fig. 4) during nighttime and daytime
sampling periods.




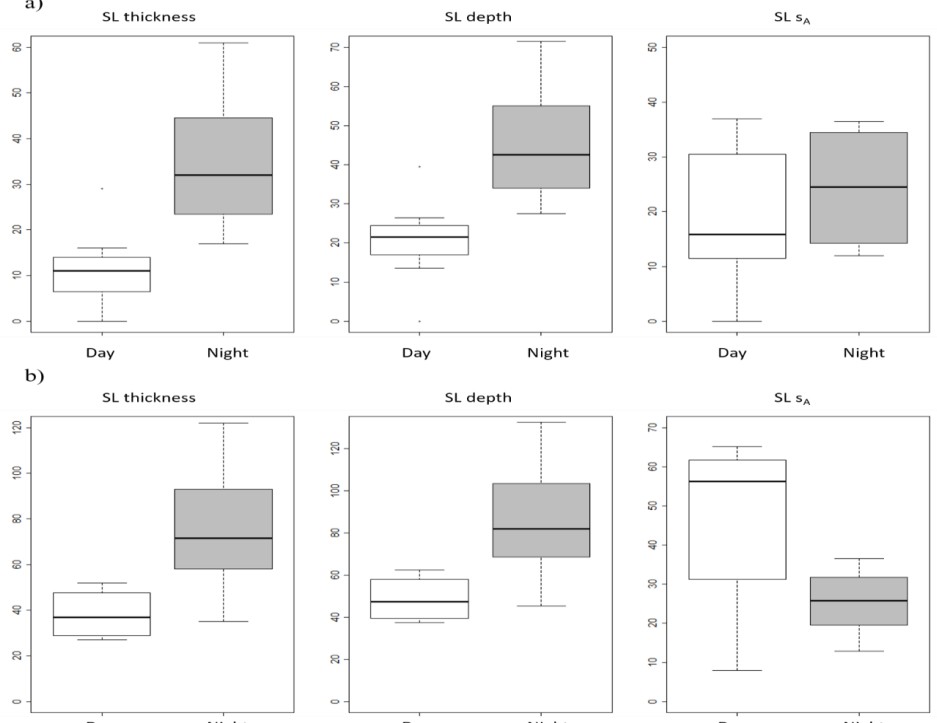

Fig. 7: Box plot (minimum, maximum, and median) of sound scattering layers (SSLs)

mean depth (m), thickness (m), and relative biomass ($s_A$ in $m^2$ $nmi^{-2}$) grouped by diel

period (days/night) for (a) inshore area (G1); and (b) offshore area (G2) over the

Senegalese continental shelf.