# Peer review of "Fine-scale vertical structure of sound scattering layers over an east border upwelling system and its relationship to pelagic habitat characteristics"

_Ocean Science, 2019_

## Referee Comment (RC1) · Dr. Heino Fock (Referee) · 19 Jun 2019

The authors present an interesting data set on coastal hydroacoustics in an African upwelling area. However, despite CTD measurements, additional measurements have not been undertaken. Thus no information on zooplankton or fish composition is made, and additional frequencies to 38kHz that could be used for a relative frequency response analysis to indicate the differential contributions of the main hydroacoustic compartments fluid-like species, fishes with swim-bladder etc. have not be sampled. One such paper is mentioned in the ref list (Behagle et al 2017). At least, the authors should

consider presenting Sv histograms where possible. Altogether, this imposes some limitations on the overall scope of the analysis. Significant clarifications should be done. In line 126, the authors indicate some relationship to ichthyoplankton, eggs and larvae, whichs needs some clarification given the properties of this features: Larvae with or without swim bladders, and how do the authors suggest fish eggs to be detected in an echogram. I know that one of the authors has undertaken significant research in indicating fish schools by shape in echograms, no mention of this work is indicated here. The authors should please consider the following 4 points in detail and improve the English: 1 Physico-chemical properties and analysis of water massess (line 135) Dissolved oxygen was measured with a sensor - was this sensor calibrated by chemical measurements? It is further mentioned, that DO concentrations did not satisfy criteria for hypoxia (line 392) as defined as 1.42 ml l-1. However, looking at pages 24/25 the DO profiles clearly indicate layers with DO concentrations of 1 ml l-1 and below, which indicate local hypoxia. Accordingly, the vertical variability of water mass properties with regards to DO and others questions the approach to cluster stations only based on their properties at 10 m water depth.

2 Definition of terms and surrogate variables and correlations with physicochemical properties (line 147) SSL thickness and maximum depth are returned by the algorithm "layer" - however, it is not mentioned, what criteria are applied to measure this or how thickness is defined in terms of sA vertical distribution. The pseudo code at least should be shown in the appendix, and the respective analytical equations/definitions should be part of Material&Methods. (see negative statement in line 457). For "ComparEchoProfil", one 0.1 nmi unit is used for calculation around the CTD cast position, but in the figures several units are shown (normally 5). Since these are not averaged in "ComparEchoProfil", so I would leave that out so that the reader is not confused by the variability. Figure legends on echogram pages 22-25 contains strong statements like "The peak of Sv match the CHL...", which need some clarification, since already in the figure for station station 12 the Sv peak is < 40 m and the CHL peak is > 40 m depth, so they do not match.

3 Model building It is unclear, how the ANCOVA models were developed - are the calculations been carried out bin-wise or averaged over station - the residual plots indicate the latter. This needs justification, line 177 indicates some kind of 'profile coupling'.

4 Features of relevance In the abstract (line 34) no significant relationship to DO is indicated, however, considerable emphasis is attributed to this feature despite being non-significant. In the first place, models for G2 indeed indicate significance of DO, so the statement in the abstract is not clear. Secondly, if non-significant terms are discussed broadly, this needs better justification.

---

## Referee Comment (RC2) · Anonymous Referee #2 · 17 Jul 2019

Diogoul et al. present an interesting survey of sound-scattering layers on the continental shelf of Senegal. They found that SSLs were thicker, deeper, and denser offshore of the upwelling front, and assessed the influence of several environmental covariates on the location and density of the layers. These results are valuable, as SSLs are ubiquitous but generally under-described features of the world's oceans, and the wide shelf off of Senegal makes this easter-boundary upwelling system different from other better-studied systems, e.g. off the west coasts of the Americas. The authors have also presented a good review of the literature, both for this system and SSLs in upwelling

systems in general.

While I believe the information presented is valuable and worth sharing, the manuscript has a number of serious issues which need to be addressed before publication. Some major points are as follows:

1. The authors do not identify the constituent animals of the SSLs. I realize this is often difficult or impossible when acoustic data are collected opportunistically, and that raw backscatter contains valuable information even when it cannot be attributed to specific animals or converted to biomass. However, it is extremely difficult to interpret these results without knowing at least broadly what kinds of animals are present. The authors need to provide more information on the likely sources of backscatter, even if it is only based on previous studies in this area.

2. Some of the methods and results need to be described in more detail. In particular, the acoustic analysis and methods used to extract and measure the scattering layers are not sufficient. The statistical models are also not described in enough detail.

3. Some of the conclusions are not supported by the data presented. For instance, the authors refer to turbulence and advective transport, which were not measured. The lack of information on the sound-scattering species also makes parts of the discussion quite vague, reading more like a broad literature review rather than a specific discussion of the physical/biological processes likely to be at play in this particular instance. One finding which needs better explanation is the apparent reverse diel migration (up during day, down at night). One possibility which needs to be considered is cross-shore diel migration (see comment and suggested reference below).

4. Several of the figures need revision (see specific comments at end).

Minor comments, questions, and line edits follow below.

47: "entire mid-trophic level" Not necessarily. There can be a large "jelly web" that is not visible to echosounders, depending on their frequency. See e.g. Choy et al.
(2017), https://doi.org/10.1098/rspb.2017.2116 72-89: This is a good overview of the literature, but I think the term "SSL" is used a little too broadly here. The authors should make it clearer that "SSL" is not a biological classification, and that animals making up SSLs can be all kinds of different things, with correspondingly different biological, physiological, and ecological needs. 90-91: Please clarify what is meant by "fine-scale" (m, 10s of m, etc.)  106: The term "radial" in this context is not standard—please replace with "transect" or "transect line" 111: What parasite? 112: "Offset" implies the data were shifted 10 m in space. I believe you mean that you discarded data from the upper 10 m? 121: Replace "extracted" with "removed." 122: A threshold of -70 dB at 38 kHz, as you say, is typical for fish surveys, but studies of smaller animals often use a threshold lower than -75 dB, which will eliminate backscatter from most zooplankton at this frequency. I'd like a little more explanation of how you selected this threshold.

Also, it appears from Figure 2 that the acoustic data were binned at coarser resolution than the raw data from the EK60. Is this the case? If so, what are the dimensions of the integrated cells?

128: Replace "security" with "safety" 141: This link is no longer functional. Are these data available somewhere else? 147-150: Please include a reference or hyperlink to the Matlab code in the body of the paper. Is there a reason it needs to be password-protected instead of, say, posted publicly on GitHub? Even though the code is shared, the algorithm you used for detecting SSLs and extracting their descriptors still needs to be described in more detail. This is a non-trivial problem and different approaches can yield different results. 163-167: Move this explanation up to come before the preceding paragraph. The goal of an analysis should be described before the software used to accomplish it. 168: Just say "Inshore-offshore variability." 169: Use past-tense verbs, i.e. "were" instead of "are" 177-180: Please describe the statistical modeling in more detail here. I found it hard to figure out exactly what was being modeled/predicted and how. 209-210: "SST revealed and advection..." It is not clear to me how SST reveals advection here.  211: Please make sure all verbs in this paragraph are past-tense.

Also, this section might read better if it was organized by zone, rather than by variable. 216: I assume these are sigma-t values? Please clarify. 233-234: I assume "depth" here refers to bottom depth? Some of the wording is a bit unclear, for instance "below 29 m depth." Does this mean "at locations seaward of the 29 m isobath," or "at locations where bottom depth was less than 29 m?" 238-240: These two sentences are confusing to me—please clarify. If the difference was not significant, why report it? 242-255: This section reads like a string of numbers. It should be revised to emphasize concise verbal descriptions of the result, with the effect sizes and p-values either incorporated into the descriptions or reported parenthetically to support the verbal descriptions. 269: "the model" there were actually several models, correct? This section needs more detail–the descriptions here and the tables to not include all the information referred to (for instance, the actual response variable and effect sizes—whose variance is being explained?) 290-331: This section contains a lot of background information which would probably fit better in the introduction. Overall, I think it could be tightened and shortened considerably. 305: "surface divergence" Was this measured, or just inferred? 333-351: I think some of these explanations need more consideration (or more explanation). Again, "SSL" is not a biological descriptor, and depending on the animals which are scattering the sound they may react in many different ways to physical forcings. 338-339: What evidence is there that rapid advection causes low residence times? Are you sure the scattering animals are even planktonic? 343-344: Different animals can respond totally differently to different physical forcings. Urmy and Horne (2016) found that SSLs moved up and probably offshore during upwelling, but in the same ecosystem SSLs can actually intensify during upwelling as krill and anchovies swarm more closely (Benoit-Bird et al. 2019, https://doi.org/10.1029/2018GL081603) 351: Couldn't the SSLs be thicker and deeper farther offshore because there is more room in the water column? 364-365: One explanation for this phenomenon might be diel horizontal migration. See for instance:

Benoit-Bird, K. J., Au, W. W. L., Brainard, R. E., and Lammers, M. O. 2001. Diel horizontal migration of the Hawaiian mesopelagic boundary community observed acoustically. Marine Ecology Progress Series, 217: 1–14.

378-379: Again, the term "SSL" is being used quite broadly here. Without some reference to the actual animals making up a particular SSL—even in broad taxonomic or trophic terms—it is hard to talk informatively about particular environmental influences. 392: If DO is relatively high everywhere, does it require this much discussion? 415: "primarily a function of temperature..." I would say light is at least as important as temperature, especially for migrating animals. 418-419: On the previous page, you said DO had only a limited influence on the SSLs' positions? 445-446: The authors did not measure either turbulence or advection, so I don't think this assertion in the concluding paragraph is supported.

General comment for all heatmap-style figures: please redo these with perceptually-uniform colormaps (such as the "parula" colormap which is now the default in Matlab). As-is, these figures will not reproduce well in black and white, and will be difficult for colorblind readers to interpret. Even in color, the uneven luminance introduces artifacts which make interpretation more difficult. For instance, in Figure 1, temperatures of 19.5, 20.5, and 22.5 all have similar brightness values, and the red-to-green color ramp in the middle of the scale will be very difficult for red-green colorblind readers to interpret.

Figure 2: How were the acoustic sections in the top panels selected? The first one is more than 12 hours long, while the second is only 3 hours. It would be helpful on both of these to show a distance scale, as well as periods of light and dark. Also, the red lines indicating the station locations are difficult to see, especially in part (a). Figure 4: This figure is a repeat of Figure 1 and can be deleted. Figure 5: It would be really valuable to add a fifth column with average backscatter to this figure. Figure 6: This figure is very difficult to read, and should be redone it in a different format. I would suggest making three subplots, one for each transect ("radial"). Each of these subplots would have bottom depth or distance offshore on the x-axis, and depth below the surface on the y-axis. Mean layer depth at each station would then be plotted as

a point, with the size of the point proportional to the layer's NASC. Error bars/whiskers above and below each point would give a visual indication of its thickness. Points could be colored differently to show if they were recorded in daytime or nighttime.
* * *

---

## Author Comment (AC1) · 10 Aug 2019

We would like to thank the referee for the detailed report, his comments were very useful for improving our manuscript. Below we provide the answers to all comments.

The authors present an interesting data set on coastal hydroacoustics in an African upwelling area. However, despite CTD measurements, additional measurements have not been undertaken. Thus no information on zooplankton or fish composition is made, and additional frequencies to 38kHz that could be used for a relative frequency re-
sponse analysis to indicate the differential contributions of the main hydroacoustic compartments fluid-like species, fishes with swim-bladder etc. have not be sampled. One such paper is mentioned in the ref list (Behagle et al 2017).

Answer : This is absolutely right. In this study, we used the acoustic monofrequency approach (using 38 kHz, one of the most current frequencies used in fisheries surveys) to study the spatio-temporal SSLs structuration in relation to the environment at generic level, i.e., without species identification. One limitation was the lack of taxonomic information about the species composition of SSLs but that will case of most part of the acoustic sea surveys available worldwide. We do not present information on species composition, because our study focus on ecosystem organisation (as often described by ecosystem modellers who unfortunately do not consider species level communities) aiming to describe the spatio-temporal SSLs structuration using only 38 kHz. The 38 kHz frequency offers the advantages of depth-penetration covering the whole vertical range of SSLs. The figure 1 allow to clearly observe that the signal is mainly associated to 38 kHz and allow to find the same SSL shape than combined frequency. Lastly, even without species composition operation (which remain in any case very punctual, subject to several bias (avoidance, differential catchability per species, etc.) and difficult to extrapolate to all the area sampled)) the understanding of the relationship between SSLs and pelagic habitat characteristics is a substantial step to understand ecosystem dynamics, e.g., such results increase the understanding of migration patterns of zooplankton and micronekton as well as will improve dispersal models for organisms in upwelling regions.

At least, the authors should consider presenting Sv histograms where possible. Altogether, this imposes some limitations on the overall scope of the analysis.

Answer : Yes, it is possible and of course more relevant to add histograms of mean Sv distribution (Fig.2) of SSLs. According to your remark these figures will be added as a new Appendix H in the revised version.

Significant clarifications should be done. In line 126, the authors indicate some relationship to ichthyoplankton, eggs and larvae, which needs some clarification given the properties of this features: Larvae with or without swim bladders, and how do the authors suggest fish eggs to be detected in an echogram. Answer : Although we did do not carry out biological sampling, we assume that the backscattering was due to zooplankton and micronekton. Indeed, literature indicate the presence of zooplankton (mostly copepods), fish larvae and eggs (Ndour et al., 2018; Tiedemann and Brehmer, 2017) in our study area. Moreover, few studies have shown that fish larvae can be investigated using acoustic techniques (e.g., Castro and Bonecker, 2017; García Seoane et al., 2016). Then, when they are present in abundance, fish larvae can contribute to backscatter (García Seoane et al., 2016). For us there is no doubt that ichtchyoplankton contribute to backscatter as they fill their swim bladder with air pretty early in their life (nevertheless there was no checking for this during the larval fish identification process but the fish larvae still were only a small fraction of the whole plankton community). Therefore, we can state that backscattering was mainly due to micronekton, and zooplankton (especially fish larvae, and copepod). In line 124-125 we do not state that the backscatter come from egg and fish larvae only but by zooplankton and micronekton. Then in the next sentence line 126 we underline the fact that high abundance of egg and fish larvae have been reported during the sea survey in this area. To avoid confusion we delete fish egg in former line 126 of the MS.

I know that one of the authors has undertaken significant research in indicating fish schools by shape in echograms, no mention of this work is indicated here. Answer : Yes, theses references (Brehmer et al., 2007, 2018) will be added on the manuscript as suggested by the referee. In the MS we add this reference as follow in the section "2.2. Data analysis":

"The output included meta information [station ID, station date, station time, latitude and longitude, diel phase (day, night), and local shelf depth (bottom depth)], all of which we associated with SSLs descriptors [SSL thickness, maximum SSL depth, mean volume

backscattering strength (Sv, dB) and the mean nautical area backscattering coefficient (sA, NASC); based on classic fish school descriptors (Brehmer et al., 2007, 2019) and physico-chemical parameters associated with each SSL."

The authors should please consider the following 4 points in detail and improve the English: - The whole manuscript polished and corrected for English. 1. Physico-chemical properties and analysis of water masses (line 135) Dissolved oxygen was measured with a sensor - was this sensor calibrated by chemical measurements?

Answer : The CTD have been sent to seabird (USA) a couple of months before the survey to be calibrated. During the survey data delivered by the Seabird SBE43 sensor for DO, data have been checked by Winkler test by a marine chemistry scientist on board, a member of the Ecoao survey crew (Francois Baurand, IRD, US-Imago). We add inside the methodological section: "The CTD have been calibrated before the survey. During the survey data delivered by the SBE43 for DO have been corrected by Winkler test."

It is further mentioned, that DO concentrations did not satisfy criteria for hypoxia (line 392) as defined as 1.42 ml l-1. However, looking at pages 24/25 the DO profiles clearly indicate layers with DO concentrations of 1 ml l-1 and below, which indicate local hypoxia. Accordingly, the vertical variability of water mass properties with regards to DO and others questions the approach to cluster stations only based on their properties at 10 m water depth.

Answer : We agree with the referee regarding his remark (at line 392, where we have mentioned that DO concentrations did not satisfy criteria for hypoxia) and thanks for this. Indeed, we were wrong with the mean value of DO in the SSL with the minimum DO value at the station. We have considered this remark and have modified this part in the text MS as reported by the referee. "Previous studies (Bertrand et al., 2010; Bianchi et al., 2013; Netburn and Koslow, 2015) have suggested that vertical distributions of SSLs organisms are limited by mid-water DO concentrations which

constraint SSLs depth. These authors found a relationship between SSLs depths and hypoxia. However, in our study, we found correlation between SSLs (depth, thickness) and DO as expected, but vertical distribution of SSLs was not constrained by DO. Despite hypoxia local condition found in some stations (DO <1.42 ml l-1), SSLs appeared; Consequently, DO was not a limiting factor".

Physico-chemical characterization of water masses by clustering approach is only based on their properties at 10 m water corresponding to the Mixed Layer Depth (MLD). This choice has been made to homogenise the data due the variability depth of each station done in shallow part of the shelf and deeper ones.

2. Definition of terms and surrogate variables and correlations with physicochemical properties (line 147) SSL thickness and maximum depth are returned by the algorithm "layer" - however, it is not mentioned, what criteria are applied to measure this or how thickness is defined in terms of sA vertical distribution. The pseudo code at least should be shown in the appendix, and the respective analytical equations/definitions should be part of Material&Methods (see negative statement in line 457).

Answer : To define the descriptors (Thickness, Depth, sA, Sv) of the SSLs, we used a threshold of -75dB, which were indicated in the algorithm "contourf.m" of Matlab (Matlab code can be found at https://ch.mathworks.com/help/matlab/ref/contourf.html). This allowed to contour (by calculation of iso-lines according to the selected Sv threshold) the attached echo groups that formed the SSLs. The echoes within each SSL were extracted, and the associated descriptors were calculated from Matecho (code already available on web see MS see section 6 "software and code availability).

The pseudo code "Layer" allows to recover echointegrated echogram and SL descriptors calculated from Matecho, showed below and added on Appendix A in the revised version. The pseudo code "ComparEchoProfil" allow to calculate mean Sv and sA profiles associated to each CTD profiles and to compare them (see below equations 1 to 3) and to extract station meta information (see code below for details). We clarify

inside the revised text of the MS: "After extracting SSLs with Matecho, we developed an ad hoc Matlab extension of Matecho named "Layer" (Appendix A) to recover SL thickness, minimum and maximum SL depths (Dmin. and Dmax., respectively) and echointergrated echogram from Matecho output files and to provide it to another Matlab program "ComparEchoProfil" (Appendix B). ComparEchoProfil allows to fit in time and depth echointegrated echograms to the associated CTD vertical profiles. We used the equation to calculate thickness and also mean sA and Sv profiles based on the average of three ESUs: the ESU nearest to the CTD position and previous and following in correspondence with CTD depths (equations are not supported in this file, see the document pdf version added in supplement) ".

For "ComparEchoProfil", one 0.1 nmi unit is used for calculation around the CTD cast position, but in the figures several units are shown (normally 5). Since these are not averaged in "ComparEchoProfil", so I would leave that out so that the reader is not confused by the variability.

Answer : As the referee noticed, it look like we displayed 5 units of ESU on CTD graph, but in fact there is three with the first and the last (ESU 1 and 5) are cut in the middle and are not taken into account in the average (it is just for full illustration "graphic output of ComparEchoProfil" of the three ESUs averaged i.e. without ESU cut). These 5 ESU are not averaged in "ComparEchoProfil", only three. Our aim was to do a fine scale analysis. This is why we have choose 0.1 nmi for ESU, the idea to display the other ESU around the CTD one is to get an idea of the variability when it is possible (this is also the reason why we put time on the x-axis and not nmi). Following referee's recommendations, we have removed the zoomed echogram (Fig.3) to avoid the reader to be confused by variability in Figure 2 and former Appendix D. We have also improved the figure by adding a distance scale showing showing diel period as requested by the referee #2.

Figure legends on echogram pages 22-25 contains strong statements like "The peak of Sv match the CHL...", which need some clarification, since already in the figure for

station 12 the Sv peak is < 40 m and the CHL peak is > 40 m depth, so they do not match.

Answer : That is true. We agree that the peak of Sv don't match accurately the CHL peak at station 12 even if it almost match (e.g. station $\Delta$ âĽ̌ 5 m) for station 13, 16 and 25. However, to avoid such claims, we will just say that: "the peak CHL is near the Sv peak (above or in the middle)". We correct the legend of the former Appendix D deleting "Match" as follow:

"Appendix D: Vertical profile from CTD stations associated to acoustic volume backscattering strength (Sv, in dB) integrated per elementary sampling unit (ESU) of 0.1 nmi for 4 stations: station 12, 13, 16 and 25. The peak of Sv is close to the fluorescence peak (proxy chlorophyll-a concentration, in $\mu$g l-1) and are related to strong gradients of water temperature, itself related to water density and dissolved oxygen. From the top left to bottom right (i) vertical profile Sv (dB) in the sound scattering layers (SSLs) ; (ii) Profile of mean temperature in SSLs (°C) ; (iii) profile of fluorescence in SSLs; (iv) profile of dissolved oxygen in SSLs ($\mu$mol kg-1); (v) and profile of water density in SSLs (kg m-3)."

3. Model building. It is unclear, how the ANCOVA models were developed - are the calculations been carried out bin-wise or averaged over station - the residual plots indicate the latter. This needs justification, line 177 indicates some kind of 'profile coupling'.

Answer : The ANCOVA model were developed on averaged data over station. At line n° 144, we refer to the Echogram- profiles CTD coupling, i.e., the Echogram-profiles comparison approach used in this study. We clarify inside the text as follow "An ANCOVA test (analysis of covariance) (Wilcox, 2017) was implemented for SSLs characteristics (thickness, depth, and density). This ANCOVA model were developed on averaged data over station. "

4. Features of relevance In the abstract (line 34) no significant relationship to DO is

indicated, however, considerable emphasis is attributed to this feature despite being non-significant. In the first place, models for G2 indeed indicate significance of DO, so the statement in the abstract is not clear. Secondly, if non-significant terms are discussed broadly, this needs better justification.

Answer : In the abstract (line 34), we discuss CHL and not DO. We believe there is a confusion between Chlorophyll and Dissolved Oxygen. Indeed, we wrote: Âń chlorophyll-a has statistically no effect on SSLs structure, we report that the chlorophyll-a peak was always located above or in the middle of the SSLs Âż. Chlorophyll-a was insignificant in all model while DO was significant in the offshore area (G2). Nevertheless, according to previous referee comments we have modified the end of the abstract as follow "Lastly, over the Senegalese continental shelf the level of dissolved oxygen was not always a limiting factor, despite local hypoxia reported below 30 m depth, for SSLs marine pelagic organisms during upwelling event". In the context of climate change it is important to check carefully the impact of DO on marine pelagic spatial organisation. Tropical Atlantic minimum oxygen zone (OMZ) already expend (see, Stramma et al works) and study on the shelf is of primary importance due to fishing activity which is particularly high in our study area and there is for instance report only from one sea survey cruise lead in 2013 (Ecoao within AwA project).

Please also note the supplement to this comment:
https://www.ocean-sci-discuss.net/os-2019-23/os-2019-23-AC1-supplement.pdf
* * *
[Figure]

**Figure 1 revision: [only for referee] ;** representation of an ECOAO echogram (from 06th -07th) at three frequencies. Each frequency (here 38, 70 and 120 kHz) was scaled from 0--255 and assigned a colour RGB value based on its frequency. This method allow to clearly observe that the signal is mainly associated to 38 kHz and allow to find the same SSL shape than combined frequency.

**Fig. 1.**

A)

B)

[Figure]

Appendix A: Mean $S_v$ distribution of SSLs during daytimes (A) and night-times (B) in the study area.

**Fig. 2.**

[Figure]

Fig. 1: Echograms and associated vertical acoustic profiles as well as physico-chemical parameters (CTD data) for two example stations: (a) station 19 in the "inshore area" and (b) station 12 in the "offshore area". For both (a) and (b), top panels are echogram data collected along the transect (nmi), whereas the bottom panels depict acoustic and environmental data (depicted by the red vertical line in top panels). Environmental data for the sound scattering layer (SSL) collected at the stations at the time depicted by dotted vertical lines. Data represent mean conditions for the station collected within an area of 0.1 nmi area around the station: acoustic volume backscattering strength (Sv) SSL, temperature profile SSL, CHL profile SSL, oxygen profile SSL, and density profile SSL.

**Fig. 3.**

**Supplement:**

**Reply to referees for the manuscript** "Fine-scale vertical structure of sound scattering layers over an east border upwelling system and its relationship to pelagic habitat characteristics" by Diogoul *et al.*

We would like to thank the referee for the detailed report, his comments were very useful for improving our manuscript. Below we provide the answers to all comments.

Our reply to the referee 1 comments are written in blue.

The authors present an interesting data set on coastal hydroacoustics in an African upwelling area. However, despite CTD measurements, additional measurements have not been undertaken. Thus no information on zooplankton or fish composition is made, and additional frequencies to 38kHz that could be used for a relative frequency response analysis to indicate the differential contributions of the main hydroacoustic compartments fluid-like species, fishes with swim-bladder etc. have not be sampled. One such paper is mentioned in the ref list (Behagle et al 2017).

**Answer** : This is absolutely right. In this study, we used the acoustic monofrequency approach (using 38 kHz, one of the most current frequencies used in fisheries surveys) to study the spatio-temporal SSLs structuration in relation to the environment at generic level, i.e., without species identification. One limitation was the lack of taxonomic information about the species composition of SSLs but that will case of most part of the acoustic sea surveys available worldwide. We do not present information on species composition, because our study focus on ecosystem organisation (as often described by ecosystem modellers who unfortunately do not consider species level communities) aiming to describe the spatio-temporal SSLs structuration using only 38 kHz. The 38 kHz frequency offers the advantages of depth-penetration covering the whole vertical range of SSLs. Lastly, even without species composition operation (which remain in any case very punctual, subject to several bias (avoidance, differential catchability per species, etc.) and difficult to extrapolate to all the area sampled)) the understanding of the relationship between SSLs and pelagic habitat characteristics is a substantial step to understand ecosystem dynamics, e.g., such results increase the understanding of migration patterns of zooplankton and micronekton as well as will improve dispersal models for organisms in upwelling regions.

[Figure]

**Figure 1 revision: [only for referee] ;** representation of an ECOAO echogram (from 06[th] -07[th]) at three frequencies. Each frequency (here 38, 70 and 120 kHz) was scaled from 0--255

and assigned a colour RGB value based on its frequency. This method allow to clearly observe that the signal is mainly associated to 38 kHz and allow to find the same SSL shape than combined frequency.

At least, the authors should consider presenting $S_v$ histograms where possible. Altogether, this imposes some limitations on the overall scope of the analysis.

**Answer** : Yes, it is possible and of course more relevant to add histograms of mean $S_v$ distribution (see below) of SSLs. According to your remark these figures will be added as a new Appendix in the revised version (see below).

Appendix H a: Mean distribution $S_v$ of SSLs during daytimes in the study area.

[Figure]

Appendix H b: Mean distribution $S_v$ of SSLs during night-times in the study area.

[Figure]

Significant clarifications should be done. In line 126, the authors indicate some relationship to ichthyoplankton, eggs and larvae, which needs some clarification given the properties of this features: Larvae with or without swim bladders, and how do the authors suggest fish eggs to be detected in an echogram.

**Answer** : Although we did do not carry out biological sampling, we assume that the backscattering was due to zooplankton and micronekton. Indeed, literature indicate the presence of zooplankton (mostly copepods), fish larvae and eggs (Ndour et al., 2018; Tiedemann and Brehmer, 2017) in our study area. Moreover, few studies have shown that fish larvae can be investigated using acoustic techniques (e.g., Castro and Bonecker, 2017; García Seoane et al., 2016). Then, when they are present in abundance, fish larvae can contribute to backscatter (García Seoane et al., 2016). For us there is no doubt that ichtchyoplankton contribute to backscatter as they fill their swim bladder with air pretty early in their life (nevertheless there was no checking for this during the larval fish identification process but the fish larvae still were only a small fraction of the whole plankton community). Therefore, we can state that backscattering was mainly due to micronekton, and zooplankton (especially fish larvae, and copepod). In line 124-125 we do not state that the backscatter come from egg and fish larvae only but by zooplankton and micronekton. Then in the next sentence line 126 we underline the fact that high abundance of egg and fish larvae have been reported during the sea survey in this area. To avoid confusion we delete fish egg in former line 126 of the MS.

I know that one of the authors has undertaken significant research in indicating fish schools by shape in echograms, no mention of this work is indicated here.

**Answer** : Yes, theses references (Brehmer et al., 2007, 2018) will be added on the manuscript as suggested by the referee. In the MS we add this reference as follow in the section "2.2. Data analysis":

"The output included meta information [station ID, station date, station time, latitude and longitude, diel phase (day, night), and local shelf depth (bottom depth)], all of which we associated with SSLs descriptors [SSL thickness, maximum SSL depth, mean volume backscattering strength ($S_v$, dB) and the mean nautical area backscattering coefficient ($s_A$, NASC); based on classic fish school descriptors (Brehmer et al., 2007, 2019) and physico-chemical parameters associated with each SSL."

The authors should please consider the following 4 points in detail and improve the English:
- The whole manuscript polished and corrected for English.
1. Physico-chemical properties and analysis of water masses (line 135) Dissolved oxygen was measured with a sensor - was this sensor calibrated by chemical measurements?

**Answer** : The CTD have been sent to seabird (USA) a couple of months before the survey to be calibrated. During the survey data delivered by the Seabird SBE43 sensor for DO, data have been checked by Winkler test by a marine chemistry scientist on board, a member of the Ecoao survey crew (Francois Baurand, IRD, US-Imago). We add inside the methodological section: "The CTD have been calibrated before the survey. During the survey data delivered by the SBE43 for DO have been corrected by Winkler test."

It is further mentioned, that DO concentrations did not satisfy criteria for hypoxia (line 392) as defined as 1.42 ml l$^{-1}$. However, looking at pages 24/25 the DO profiles clearly indicate layers with DO concentrations of 1 ml l$^{-1}$ and below, which indicate local hypoxia. Accordingly, the vertical variability of water mass properties with regards to DO and others questions the approach to cluster stations only based on their properties at 10 m water depth.

**Answer** : We agree with the referee regarding his remark (at line 392, where we have mentioned that DO concentrations did not satisfy criteria for hypoxia) and thanks for this. Indeed, we were wrong with the mean value of DO in the SSL with the minimum DO value at the station. We have considered this remark and have modified this part in the text MS as reported by the referee.

"Previous studies (Bertrand et al., 2010; Bianchi et al., 2013; Netburn and Koslow, 2015) have suggested that vertical distributions of SSLs organisms are limited by mid-water DO concentrations which constraint SSLs depth. These authors found a relationship between SSLs depths and hypoxia. However, in our study, we found correlation between SSLs (depth, thickness) and DO as expected, but vertical distribution of SSLs was not constrained by DO. Despite hypoxia local condition found in some stations (DO <1.42 ml l$^{-1}$), SSLs appeared; Consequently, DO was not a limiting factor".

Physico-chemical characterization of water masses by clustering approach is only based on their properties at 10 m water corresponding to the Mixed Layer Depth (MLD). This choice has been made to homogenise the data due the variability depth of each station done in shallow part of the shelf and deeper ones.

2. Definition of terms and surrogate variables and correlations with physicochemical properties (line 147) SSL thickness and maximum depth are returned by the algorithm "layer" - however, it is not mentioned, what criteria are applied to measure this or how thickness is defined in terms of $s_A$ vertical distribution. The pseudo code at least should be shown in the appendix, and the respective analytical equations/definitions should be part of Material&Methods (see negative statement in line 457).

**Answer** : To define the descriptors (Thickness, Depth, $s_A$, $S_v$) of the SSLs, we used a threshold of -75dB, which were indicated in the algorithm "contourf.m" of Matlab (Matlab code can be

found at https://ch.mathworks.com/help/matlab/ref/contourf.html). This allowed to contour (by calculation of iso-lines according to the selected $S_v$ threshold) the attached echo groups that formed the SSLs. The echoes within each SSL were extracted, and the associated descriptors were calculated from Matecho (code already available on web see MS see section 6 "software and code availability).

The pseudo code "Layer" allows to recover echointegrated echogram and SL descriptors calculated from Matecho, showed below and added on Appendix A in the revised version. The pseudo code "ComparEchoProfil" allow to calculate mean $S_v$ and $s_A$ profiles associated to each CTD profiles and to compare them (see below equations 1 to 3) and to extract station meta information (see code below for details).

We clarify inside the revised text of the MS:

"After extracting SSLs with Matecho, we developed an *ad hoc* Matlab extension of Matecho named "Layer" (**Erreur ! Source du renvoi introuvable.**) to recover SL thickness, minimum and maximum SL depths ($D_{min.}$ and $D_{max.}$, respectively) and echointergrated echogram from Matecho output files and to provide it to another Matlab program "ComparEchoProfil" (**Erreur ! Source du renvoi introuvable.**). ComparEchoProfil allows to fit in time and depth echointegrated echograms to the associated CTD vertical profiles. We used the equation below to calculate thickness:

$$\text{Thickness} = D_{max.} - D_{min.} \qquad (1)$$

and also mean $s_A$ and $S_v$ profiles based on the average of three ESUs: the ESU nearest to the CTD position ($ESU_{ctd}$) and previous and following in correspondence with CTD depths ($d_n$) :

$$\overline{s_A(d_n)} = \left. \sum_{i=ESU_{ctd}-1}^{i=ESU_{ctd}+1} s_A(i,d_n) \right/ 3 \qquad (2)$$

$$\overline{S_v(d_n)} = 10 \times \log_{10}\left( \left. \sum_{i=ESU_{ctd}-1}^{i=ESU_{ctd}+1} 10^{(S_v(i,d_n)/10)} \right/ 3 \right) \qquad (3)$$

".

**%% Layer**
clear all; %close all;
%***********************************************************************
*
% CHOIX PARAMETRES
% adresse du répertoire contenant les fichiers Echointegration.mat et Layer.mat
adress_acou='E:\ECOAO2013\Cruise_ECOAO2013\Treatment20171021_120009\CleanResults\Echointegration\';

% adresse du répertoire où sauver les résultats
adress_save='C:\Users\perroty\Documents\DEVELOPPEMENTS\TOOLS_IRD\Profil_station_ECOAOetAWA\ComparEchoProfil_Matecho\';

% indice de la frequence qui sont rangées dans l'ordre croissant de fréquence (le 38kHz est kfreq=1 pour ECOAO et kfreq=2 pour Awa)
kfreq=1;

%***********************************************************************
*

```
load([adress_acou,'Echointegration.mat'],'Time','Sv_surface','Sa_surface','depth_surface','dept
h_bottom','TransducerDepth','Night1Sunrise2Day3Sunset4','FrequencySort','BottomShift');
load([adress_acou,'Layer.mat'],'CleanLayMask','LayDescription38','LayDescriptionHeader');
LayDescription=LayDescription38;

nbcouche=size(LayDescription,1);  % LayDescription38 --> nb couche * nbre descripteur
nbesu=size(Time,2);
IdCouche=LayDescription(:,1);
IdStartCouche=LayDescription(:,5); IdEndCouche=LayDescription(:,6);
TimeStartCouche=LayDescription(:,9); TimeEndCouche=LayDescription(:,10);
DepStartCouche=LayDescription(:,11); DepEndCouche=LayDescription(:,12);
Zone=LayDescription(:,2);
EpCouche=DepEndCouche-DepStartCouche;
d=depth_surface;
nbzone=max(Zone);
EpAllZone=zeros(1,nbesu); SvAllZone=NaN(1,nbesu); SaAllZone=NaN(1,nbesu);
IndiceCoucheAllZone=NaN(1,nbesu);
DepthDebutAllZone=NaN(1,nbesu); DepthFinAllZone=NaN(1,nbesu);
IdDepthDebutAllZone=NaN(1,nbesu); IdDepthFinAllZone=NaN(1,nbesu);
for izone=1:nbzone
    tmp0=find(Zone==izone);
    if(~isempty(tmp0))

        tmp0=tmp0(1); ZoneId=[IdStartCouche(tmp0):IdEndCouche(tmp0)];

        layer=CleanLayMask(:,ZoneId,kfreq); Sv0=Sv_surface(:,ZoneId,kfreq);
Sa0=Sa_surface(:,ZoneId,kfreq); Id0=NaN(1,length(ZoneId));

        Ep=zeros(1,length(ZoneId)); Sv=NaN(1,length(ZoneId)); Sa=NaN(1,length(ZoneId));
        DepthDebut=NaN(1,length(ZoneId)); DepthFin=NaN(1,length(ZoneId));
IdDepthDebut=NaN(1,length(ZoneId)); IdDepthFin=NaN(1,length(ZoneId));
        for k=1:length(ZoneId)
            tmp=find(layer(:,k)~=0);
            if(~isempty(tmp))
                IdDebut=min(tmp); IdFin=max(tmp); Id0(k)=layer(IdDebut,k);
                if(layer(IdDebut,k)==layer(IdFin,k))
                    Sv(1,k)=10.*log10(nanmean(10.^(Sv0(IdDebut:IdFin,k)./10)));
                    Sa(1,k)=nanmean(Sa0(IdDebut:IdFin,k));
                    Ep(1,k)=d(IdFin)-d(IdDebut);
                    DepthDebut(1,k)=d(IdDebut); DepthFin(1,k)=d(IdFin);
IdDepthDebut(1,k)=IdDebut; IdDepthFin(1,k)=IdFin;
                    if(Sv(1,k)==0)
                        Sv(1,k)=NaN; Sa(1,k)=NaN;
                    end
                else
                    tmp2=find(layer(:,k)~=layer(IdDebut,k)); layer2=layer(:,k); layer2(tmp2)=0;
clear tmp2;
                    clear tmp; tmp=find(layer2~=0);
                    if(~isempty(tmp))
                        Sv(1,k)=10.*log10(nanmean(10.^(Sv0(IdDebut:IdFin,k)./10)));
```

```matlab
                Sa(1,k)=nanmean(Sa0(IdDebut:IdFin,k));
                Ep(1,k)=d(max(tmp))-d(min(tmp));
                DepthDebut(1,k)=d(IdDebut); DepthFin(1,k)=d(IdFin);
IdDepthDebut(1,k)=IdDebut; IdDepthFin(1,k)=IdFin;
                if(Sv(1,k)==0)
                    Sv(1,k)=NaN; Sa(1,k)=NaN;
                end
            end
            clear layer2;
        end
    end
    clear tmp;
end
IndiceCoucheAllZone(1,ZoneId(1):ZoneId(end))=Id0;
EpAllZone(1,ZoneId(1):ZoneId(end))=Ep;
SvAllZone(1,ZoneId(1):ZoneId(end))=Sv;
SaAllZone(1,ZoneId(1):ZoneId(end))=Sa;
DepthDebutAllZone(1,ZoneId(1):ZoneId(end))=DepthDebut;
DepthFinAllZone(1,ZoneId(1):ZoneId(end))=DepthFin;
IdDepthDebutAllZone(1,ZoneId(1):ZoneId(end))=IdDepthDebut;
IdDepthFinAllZone(1,ZoneId(1):ZoneId(end))=IdDepthFin;

    clear ZoneId layer Ep Sv Sa Id0 DepthDebut DepthFin IdDepthDebut IdDepthFin;
  end
  clear tmp0;
end
save([adress_save,'EpSvSa.mat'],'IndiceCoucheAllZone','EpAllZone','SvAllZone','SaAllZone'
,'DepthDebutAllZone','DepthFinAllZone','IdDepthDebutAllZone','IdDepthFinAllZone')
hf=figure; subplot(6,1,1); plot(IndiceCoucheAllZone); title('Indice de la couche sur laquelle
epaisseur est estime');
subplot(6,1,2); plot(DepthDebutAllZone); title('Profondeur minimale');
subplot(6,1,3); plot(DepthFinAllZone); title('Profondeur maximale');
subplot(6,1,4); plot(EpAllZone); title('Epaisseur');
subplot(6,1,5); plot(SvAllZone); title ('Sv'); ylim([-80 -20]);
subplot(6,1,6); plot(SaAllZone); title('Sa'); ylim([0 800]);
```

%% ComparEchoProfil

clear all; close all; fclose all; warning('off');

%================================================================
=========
% ECOAO : type de profils
% # name 1 = prDM: Pressure, Digiquartz [db]
% # name 2 = t090C: Temperature [ITS-90, deg C]
% # name 3 = t190C: Temperature, 2 [ITS-90, deg C]
% # name 4 = c0S/m: Conductivity [S/m]
% # name 5 = c1S/m: Conductivity, 2 [S/m]
% # name 6 = sbeox0V: Oxygen raw, SBE 43 [V]
% # name 7 = sbeox1V: Oxygen raw, SBE 43, 2 [V]
% # name 8 = par: PAR/Irradiance, Biospherical/Licor
% # name 9 = spar: SPAR/Surface Irradiance
% # name 10 = flC: Fluorescence, Chelsea Aqua 3 Chl Con [ug/l]
% # name 11 = 100-CStarTr0: Beam Transmission, WET Labs C-Star [%]
% # name 12 = altM: Altimeter [m]
% # name 13 = sbeox0ML/L: Oxygen, SBE 43 [ml/l], WS = 2
% # name 14 = sbeox0Mm/Kg: Oxygen, SBE 43 [umol/Kg], WS = 2
% # name 15 = sbeox1ML/L: Oxygen, SBE 43, 2 [ml/l], WS = 2
% # name 16 = sbeox1Mm/Kg: Oxygen, SBE 43, 2 [umol/Kg], WS = 2
% # name 17 = nbin: number of scans per bin
% # name 18 = sal00: Salinity, Practical [PSU]
% # name 19 = sal11: Salinity, Practical, 2 [PSU]
% # name 20 = sigma-é00: Density [sigma-theta, Kg/m^3]
% # name 21 = svCM: Sound Velocity [Chen-Millero, m/s]
%================================================================
=========
% AWA2014 : DESCRPITION OF THE PARAMETERS IN MATRIX "PROF", SAVED IN
EACH MATRIX profil_d_**.mat FOR DOWN PROFILS and profil_u_**.mat FOR UP
PROFILS
% # name 1 = timeJ: Julian Days
% # name 2 = prDM: Pressure, Digiquartz [db]
% # name 3 = t090C: Temperature [ITS-90, deg C]
% # name 4 = t190C: Temperature, 2 [ITS-90, deg C]
% # name 5 = c0S/m: Conductivity [S/m]
% # name 6 = c1S/m: Conductivity, 2 [S/m]
% # name 7 = sbeox0V: Oxygen raw, SBE 43 [V]
% # name 8 = sbeox1V: Oxygen raw, SBE 43, 2 [V]
% # name 9 = flC: Fluorescence, Chelsea Aqua 3 Chl Con [ug/l]
% # name 10 = CStarTr0: Beam Transmission, WET Labs C-Star [%]
% # name 11 = nbf: Bottles Fired
% # name 12 = sbeox0Mm/Kg: Oxygen, SBE 43 [umol/kg]
% # name 13 = sbeox1Mm/Kg: Oxygen, SBE 43, 2 [umol/kg]
% # name 14 = sal00: Salinity, Practical [PSU]
% # name 15 = sal11: Salinity, Practical, 2 [PSU]
% # name 16 = sigma-é00: Density [sigma-theta, kg/m^3]
% # name 17 = sigma-é11: Density, 2 [sigma-theta, kg/m^3]

```matlab
% # name 18 = density00: Density [density, kg/m^3]
% # name 19 = density11: Density, 2 [density, kg/m^3]
% # name 20 = svCM: Sound Velocity [Chen-Millero, m/s]
% # name 21 = svCM1: Sound Velocity, 2 [Chen-Millero, m/s]
% # name 22 = nbin: Scans Per Bin
%=================================================================
=========
prompt = {'CAMPAGNE (taper ECOAO ou AWA)','CHOIX DE LA FREQUENCE A
ANALYSER (ECOAO--> 38, 70, 120 ou 200 kHz - AWA--> 18, 38, 70, 120, 200 ou 333
kHz)'};
dlg_title = 'Comparaison profils CTD et Echogrammes'; num_lines = 1; def={'ECOAO','38'};
answer = inputdlg(prompt,dlg_title,num_lines,def,'on');
if(strcmp(char(answer(1)),'ECOAO'))
    camp=1;
else
    camp=2;
end
FREQUENCES=str2num(char(answer(2)));
%=================================================================
=========
% PARAMETRES AVANCEES

% adresse du répertoire où se trouve les données acoustiques de la campagne ECOAO et
AWA (contenant tous les fichiers du type AWA2014__Y2014M02-the16at154403-
the01at075520.mat)
%adress_acou_ECOAO='C:\Users\USER\Desktop\HacTest\N058-S014-
S1999404\Cruise_1999404\Treatment20170818_124508\CleanResults\Echointegration\';
adress_acou_ECOAO='E:\ECOAO2013\Cruise_ECOAO2013\Treatment20171021_120009\
CleanResults\Echointegration\';
adress_acou_AWA='E:\AWA2014\Cruise_AWA2014\Treatment20170615_133808\CleanRe
sults\Echointegration\';

% adresse du répertoire où se trouve les fichiers des profils (qui doit être different de
adress_acou)
adress_profil='E:\CEprofil\Station_ECOAO-AWA\';

% adresse du fichier où sont enregistrées les épaisseurs (sortie du programme "couche.m")
adress_EpSvSa='C:\Users\perroty\Documents\DEVELOPPEMENTS\TOOLS_IRD\Profil_sta
tion_ECOAOetAWA\ComparEchoProfil_Matecho\EpSvSa.mat';

NbESUVisu=10;  % nombre d'esu à visualiser autour de la station (moyenne echogramme fait
sur ce nombre d'ESU)

if(camp==1)
    TrialName='ECOAO';
    ProfilType=[2,10,14,20];
    FileNameProfil='CTD_stations_ECOAO';
    adress_acou=adress_acou_ECOAO;
else
    TrialName='AWA';
```

```
    ProfilType=[3,9,12,16];
    FileNameProfil='CTD_stations_AWA';
    adress_acou=adress_acou_AWA;
end
TypeEi='v';                              % ='v' pour analyser l'échogramme des Sv, ='a'
pour les echogrammes Sa
ProfName={'Temperature','Fluorescence','Oxygen','Density'};
ProfUnit= {'°C','µg/l','µmol/kg','kg/m^3'};
ProfilUpDown='d';                              % ='d' pour analyser le profil descendant ou
='u' pour le profil montant

% Sauvegarde des figures
SaveFIG=1;      % =1 pour sauver les figures au format matlab (il suffit de cliquer dessus
ensuite pour les ouvrir, faire des zooms, etc..)
SaveFIGppt=1;     % =1 pour sauver toutes les figures produites dans un powerpoint (très
utile)
LabelFigHH=1;     % =1 pour afficher les numéros d'ESU en heure minute (HH:MN), =0
sinon

%************************************************************************
*
% Debut du programme
%************************************************************************
*
temp=pwd;
if(~exist([temp,'\RESULTATS\'],'dir'))
    mkdir(temp,'RESULTATS');
end
pathSaveFig=[temp,'\','RESULTATS\']; clear temp;

% chargement profils
load('EK500_colourmap.dat'); ek5=EK500_colourmap; clear EK500_colourmap;
load([adress_profil,FileNameProfil,'.mat']); month=char(monthtr);

% chargement des sorties de couche.m (début et fin de couche)
load(adress_EpSvSa);

%for k=12:12
for k=1:length(hourtr)
    if(month(k,1)=='F')
        m=2;
    else
        m=3;
    end
    hh=str2num(hourtr(k,1:2)); mn=str2num(hourtr(k,4:5)); ss=str2num(hourtr(k,7:8));
    timep(k)=datenum([str2num(char(yeartr(k))),m,daytr(k),hh,mn,ss]); clear hh mn ss;
    timep70(k)=(timep(k)-datenum('1970-01-01 00:00'))*60*60*24;
    timpstr(k,:)=datestr(datenum([1970 1 1 00 00 timep70(k)]),'yyyy-mm-dd HH:MM:SS');
    if(camp==1)
```

```matlab
        temp=char(lattr(k)); ii=strfind(temp,'°'); dd=str2num(temp(1:ii-1));
mn=str2num(temp(ii+1:end-1)); lat(k)=dd + mn/60;
        if(temp(end)=='S')
            lat(k)=-lat(k);
        end
        clear temp ii dd mn;
        temp=char(lontr(k)); ii=strfind(temp,'°'); dd=str2num(temp(1:ii-1));
mn=str2num(temp(ii+1:end-1)); lon(k)=dd + mn/60;
        if(temp(end)=='W')
            lon(k)=-lon(k);
        end
        clear temp ii dd mn;
    else
        lat=lattr; lon=lontr;
    end
end
DateStation=timpstr;

filemat='Echointegration.mat';
str_fr0='000'; str_fr0(end-
length(num2str(FREQUENCES))+1:end)=num2str(FREQUENCES);
repsave=[str_fr0,'kHz_',TrialName,'_',datestr(clock,'dd-mm-yyyy_HH-MM-SS')];
mkdir([pathSaveFig,repsave]);
save([pathSaveFig,repsave,'\ParametresDeTraitement.mat'],'DateStation','TrialName','adress_
acou','adress_profil','TypeEi','FREQUENCES','FileNameProfil','ProfilType','ProfilUpDown','
ProfName');

Kstation=0; Station=[]; kfile=1; load([adress_acou,filemat(kfile,:)],'Time');
nbesutot=size(Time,2); crit=0;
NbEsuBloc=1000; IdP=[1:NbEsuBloc];
while(crit==0)
    if(IdP(end)>=nbesutot)
        IdP=[IdP(1):nbesutot];
        crit=1;
    end

load([adress_acou,filemat(kfile,:)],'Sv_surface','Sa_surface','Time','depth_surface','depth_bott
om','TransducerDepth','Night1Sunrise2Day3Sunset4','FrequencySort','BottomShift');
    load([adress_acou,'Layer.mat'],'CleanLayMask');

    fprintf('OK\n');
    Sv_surface=Sv_surface(:,IdP,:); Sa_surface=Sa_surface(:,IdP,:); Time=Time(1,IdP);
depth_bottom=depth_bottom(1,IdP);
    day_night_twilight=Night1Sunrise2Day3Sunset4(1,IdP);

    % mettre la ligne ci-dessous en commentaire (%) pour mettre tout l'échograme
    CleanLayMask=CleanLayMask(:,IdP,:); tmp=find(CleanLayMask~=0);
CleanLayMask(tmp)=1; clear tmp; tmp=find(CleanLayMask==0);
```

```matlab
CleanLayMask(tmp)=NaN; clear tmp; Sv_surface=Sv_surface.*CleanLayMask;
Sa_surface=Sa_surface.*CleanLayMask;

  transFreq=FrequencySort;
  ind=find(timep70>=Time(1) & timep70<=Time(end));
  if(~isempty(ind))

    for kp=1:length(ind)

      Kstation=Kstation+1;  % compteur de stations

      tmp=find(Time>=timep70(ind(kp))); IndStation=tmp(1);
      if(day_night_twilight(IndStation)==1)
        JourNuitStation(Kstation,:)='Nuit  ';
      elseif(day_night_twilight(IndStation)==2)
        JourNuitStation(Kstation,:)='Levé  ';
      elseif(day_night_twilight(IndStation)==3)
        JourNuitStation(Kstation,:)='Jour  ';
      elseif(day_night_twilight(IndStation)==4)
        JourNuitStation(Kstation,:)='Couché';
      end

      tempek=[IndStation-NbESUVisu:1:IndStation+NbESUVisu]; clear tmp;
      if(tempek(1)<1)
        tempek=1:1:2*NbESUVisu;
      end
      if(tempek(end)>length(Time))
        tempek=length(Time)-2*NbESUVisu:1:length(Time);
      end

      dtim=Time(tempek);
      for k=1:length(Time)
        timtot(k,:)=datestr(datenum([1970 1 1 00 00 Time(k)]),'yyyy-mm-dd HH:MM:SS');
      end
      tim=timtot(tempek,:); DateESU=timtot;
      pingdeb=find(dtim>=timep70(ind(kp)));
      if(isempty(pingdeb))
        pingdeb=find(abs(dtim-timep70(ind(kp)))==min(abs(dtim-timep70(ind(kp)))));
      end
      pingdeb=pingdeb(1);
      pingdebtot=find(Time>=timep70(ind(kp))); pingdebtot=pingdebtot(1);

      if(ProfilUpDown(1)=='d')
        Name=['profil_d_',num2str(ind(kp))]; proftot=eval(Name); clear Name;
Name=['profildepth_d_',num2str(ind(kp))]; D=eval(Name); clear Name;
      else
        Name=['profil_u_',num2str(ind(kp))]; proftot=eval(Name); clear Name;
Name=['profildepth_u_',num2str(ind(kp))]; D=eval(Name); clear Name;
      end
      prof=proftot(:,ProfilType);
```

```matlab
    for Kfreq=1:length(FREQUENCES)

        kfreq=find(transFreq==FREQUENCES(Kfreq)*1000);

        Sv=Sv_surface(:,tempek,kfreq); Svtot=Sv_surface(:,:,kfreq);
Sa=Sa_surface(:,tempek,kfreq);

        % suppression des NaNs
        Ks=1;
        for ks=1:size(Svtot,2)
            tmp=find(~isnan(Svtot(:,ks)));
            if(~isempty(tmp))
                maxind(Ks)=tmp(end)+1; Ks=Ks+1;
            end
            clear tmp;
        end
        clear Ks; maxind=max(maxind); if(maxind>size(Svtot,1)); maxind=size(Svtot,1);
end; Svtot=Svtot(1:maxind,:);

        depth=depth_surface(1,1:maxind);
        bottomtot=depth_bottom(1,:,kfreq);
        tmp=find(bottomtot>max(depth)); if(~isempty(tmp));
bottomtot(tmp)=max(depth).*ones(length(tmp),1); end; clear tmp;
        bottom=bottomtot(tempek);

        % calcul profil acoustic moyen
        Saprof=nanmean(Sa.'); Svprof=10.*log10(nanmean(10.^(Sv.'/10)));

        %==========================================================
        % FIGURE
        SupTitle='';%['PRESSE "ESC" pour faire des zooms - PRESSE "Y" pour revenir à
toute l''image - ECHELLE DE COULEUR: PRESSE "A" ou "Q" pour augmenter ou diminuer
la valeur minimale des couleurs (PRESSE "Z" ou "S" pour augmenter ou diminuer sa valeur
maximale)' ];
        limax=[1 size(Svtot,2) min(depth) max(bottomtot)]; col=[-100 -40];
zoomcurrent=limax; ClimCurrent=col; DClim=10;
        tit={''};
        titcurrent=tit; Val=0; kcount=1; X=[]; Y=[]; aff=0;

    % while(Val==0)% | Val==3)

        close all; figure('units','normalized','outerposition',[0 0 1
1],'Name',SupTitle,'NumberTitle','off');
        Y(1)=DepthDebutAllZone(IdP(1)+IndStation-1);
        Y(2)=DepthFinAllZone(IdP(1)+IndStation-1);
        X(1)=IndStation;
        X(2)=IndStation;
        Val=1;
```

```matlab
        FigManage;

    % end  %while(Val==0 | Val==3)

    if(~isnan(Y(1)))
        tmp=find(D>=Y(1) & D<=Y(2));
        if(isempty(tmp))
            Temp=NaN; Fluo=NaN; Oxy=NaN; Dens=NaN; Pc=0;
        else
            Temp=nanmean(prof(tmp,1)); Fluo=nanmean(prof(tmp,2));
Oxy=nanmean(prof(tmp,3)); Dens=nanmean(prof(tmp,4)); Pc=length(tmp)/length(D)*100;
        end
        clear tmp;

        tmp=find(depth>=Y(1) & depth<=Y(2)); Samoy=nanmean(Saprof(tmp));
tmp2=nanmean(10.^(Svprof(tmp)/10));
        if(~isnan(tmp2) & tmp2>0)
            Smoy=10.*log10(tmp2);
        else
            Smoy=NaN;
        end
        clear tmp tmp2;

Station(Kstation,:)=[X(1),X(2),D(1),D(end),depth(1),bottomtot(IndStation),Time(X(1)),Y(1),
bottomtot(X(1)),Time(X(2)),Y(2),bottomtot(X(2)),abs(Y(2)-
Y(1)),Temp,Fluo,Oxy,Dens,Smoy,Samoy,Pc];
        else % si pas de couche
            Temp=nanmean(prof(:,1)); Fluo=nanmean(prof(:,2)); Oxy=nanmean(prof(:,3));
Dens=nanmean(prof(:,4)); Smoy=10.*log10(nanmean(10.^(Svprof/10)));
Samoy=nanmean(Saprof);

Station(Kstation,:)=[NaN,NaN,D(1),D(end),depth(1),bottomtot(IndStation),NaN,NaN,NaN,N
aN,NaN,NaN,NaN,Temp,Fluo,Oxy,Dens,Smoy,Samoy,NaN];
        end
        SvMoyStation(Kstation).Sv=Svprof; SvMoyStation(Kstation).Sa=Saprof;

        %============================================================
        % SAUVEGARDES

        ClimCurrent=col; zoomcurrent=limax; aff=1; FigManage; aff=0;
        if(SaveFIG==1)
            str_st='000'; str_st(end-length(num2str(ind(kp)))+1:end)=num2str(ind(kp));
str_fr='000'; str_fr(end-
length(num2str(transFreq(kfreq)/1000))+1:end)=num2str(transFreq(kfreq)/1000);
            saveas(gcf,[pathSaveFig,repsave,'\',TrialName,'_Station',str_st,'-
',str_fr,'kHz'],'fig');
        end
        if(SaveFIGppt==1)
            saveppt([pathSaveFig,repsave,'\',TrialName,'_AllStations.ppt']);
```

```
        end

        close all; pause(0.01); clear Sv depth bottom Svtot depth bottomtot;

    end %for kfreq=1:length(transFreq)
    clear Temperature Fluorescence Oxygene Density Smoy X Y Pc; clear tempek dtim
tim proftot prof D Svprof Saprof pingdeb pingdebtot IndStation;

    end %for kp=1:length(ind)

  end % if(~isempty(ind))
  clear ind;
  IdP=IdP+NbEsuBloc;

end % for kfile=1:size(filemat,1)

if(~isempty(Station))
  LatitudeStation=lat; LongitudeStation=lon;

save([pathSaveFig,repsave,'\Stations_',TrialName,'_',str_fr0,'kHz.mat'],'JourNuitStation','Date
ESU','LatitudeStation','LongitudeStation','Station','SvMoyStation');

  % ecriture du fichier excel de résultats
  fid=fopen([pathSaveFig,repsave,'\Stations_',TrialName,'_',str_fr0,'kHz.xls'],'wt');
  fprintf(fid,'N°station \t Date station \t Latitude station (deg) \t Longitude station (deg) \t
Ephéméride à la station \t Profondeur minimale station (m) \t Profondeur maximale station
(m) \t Profondeur minimale echogramme sur station (m) \t Profondeur maximale echogramme
sur station (m) \t Date point1 (seconde depuis 1970) \t Profondeur point1 (m) \t Fond au
point1 (m) \t Date point2 (seconde depuis 1970) \t Profondeur point2 (m) \t Fond au point2
(m) \t Epaisseur couche (m) \t Temperature moyenne dans la couche (°C) \t Fluorescence
moyenne dans la couche (µg/l) \t Oxygène moyen dans la couche (µmol/kg) \t Densité
moyenne dans la couche (kg/m3) \t Sv moyen dans la couche (dB) \t Sa moyen dans la couche
(NASC) \t Pourcentage de recouvrement des profils de station par la couche\n');
  for k=1:size(Station,1)
    fprintf(fid,'%i \t %s \t %4.4f \t %4.4f \t %s \t %4.2f \t %4.2f \t %4.2f \t %4.2f \t %4.4f \t
%4.2f \t %4.2f \t %4.4f \t %4.2f \t %4.2f \t %4.2f \t %4.2f \t %4.2f \t %4.2f \t %4.2f \t %4.2f
\t %4.2f \t %4.2f
\n',k,char(timpstr(k,:)),lat(k),lon(k),char(JourNuitStation(k,:)),Station(k,3:end));
  end
  fclose all;
  fprintf('\n'); fprintf('Les resultats sont sauvegardés dans le répertoire:\n');
fprintf('%s\n',[pathSaveFig,repsave]);
  cd([pathSaveFig,repsave]);
else
  fprintf('Aucune station n"a été traitée !!!\n');
end
warning('on')"
```

For "ComparEchoProfil", one 0.1 nmi unit is used for calculation around the CTD cast position, but in the figures several units are shown (normally 5). Since these are not averaged in "ComparEchoProfil", so I would leave that out so that the reader is not confused by the variability.

**Answer** : As the referee noticed, it look like we displayed 5 units of ESU on CTD graph, but in fact there is three with the first and the last (ESU 1 and 5) are cut in the middle and are not taken into account in the average (it is just for full illustration "graphic output of ComparEchoProfil" of the three ESUs averaged i.e. without ESU cut). These 5 ESU are not averaged in "ComparEchoProfil", only three. Our aim was to do a fine scale analysis. This is why we have choose 0.1 nmi for ESU, the idea to display the other ESU around the CTD one is to get an idea of the variability when it is possible (this is also the reason why we put time on the x-axis and not nmi). Following referee's recommendations, we have removed the zoomed echogram (see below) to avoid the reader to be confused by variability in Figure 2 and former Appendix D. We have also improved the figure by adding a distance scale showing showing diel period as requested by the referee #2.

[Figure]

Figure legends on echogram pages 22-25 contains strong statements like "The peak of $S_v$ match the CHL…", which need some clarification, since already in the figure for station 12 the $S_v$ peak is < 40 m and the CHL peak is > 40 m depth, so they do not match.

**Answer** : That is true. We agree that the peak of $S_v$ don't match accurately the CHL peak at station 12 even if it almost match (e.g. station $\Delta \cong 5$ m) for station 13, 16 and 25. However, to avoid such claims, we will just say that: "the peak CHL is near the $S_v$ peak (above or in the middle)".

We correct the legend of the former Appendix D deleting "Match" as follow:

"Appendix D: Vertical profile from CTD stations associated to acoustic volume backscattering strength ($S_v$, in dB) integrated per elementary sampling unit (ESU) of 0.1 nmi for 4 stations: station 12, 13, 16 and 25. The peak of $S_v$ is close to the fluorescence peak (proxy chlorophyll-*a* concentration, in µg l$^{-1}$) and are related to strong gradients of water temperature, itself related

to water density and dissolved oxygen. From the top left to bottom right (i) vertical profile $S_v$ (dB) in the sound scattering layers (SSLs) ; (ii) Profile of mean temperature in SSLs (°C) ; (iii) profile of fluorescence in SSLs; (iv) profile of dissolved oxygen in SSLs (µmol kg$^{-1}$); (v) and profile of water density in SSLs (kg m$^{-3}$).”

3. Model building. It is unclear, how the ANCOVA models were developed - are the calculations been carried out bin-wise or averaged over station - the residual plots indicate the latter. This needs justification, line 177 indicates some kind of 'profile coupling'.
**Answer** : The ANCOVA model were developed on averaged data over station. At line n° 144, we refer to the Echogram- profiles CTD coupling, i.e., the Echogram-profiles comparison approach used in this study. We clarify inside the text as follow "An ANCOVA test (analysis of covariance) (Wilcox, 2017) was implemented for SSLs characteristics (thickness, depth, and density). This ANCOVA model were developed on averaged data over station. "

4. Features of relevance In the abstract (line 34) no significant relationship to DO is indicated, however, considerable emphasis is attributed to this feature despite being non-significant. In the first place, models for G2 indeed indicate significance of DO, so the statement in the abstract is not clear. Secondly, if non-significant terms are discussed broadly, this needs better justification.
**Answer** : In the abstract (line 34), we discuss CHL and not DO. We believe there is a confusion between Chlorophyll and Dissolved Oxygen. Indeed, we wrote: « chlorophyll-*a* has statistically no effect on SSLs structure, we report that the chlorophyll-*a* peak was always located above or in the middle of the SSLs ». Chlorophyll-*a* was insignificant in all model while DO was significant in the offshore area (G2). Nevertheless, according to previous referee comments we have modified the end of the abstract as follow "Lastly, over the Senegalese continental shelf the level of dissolved oxygen was not always a limiting factor, despite local hypoxia reported below 30 m depth, for SSLs marine pelagic organisms during upwelling event".
In the context of climate change it is important to check carefully the impact of DO on marine pelagic spatial organisation. Tropical Atlantic minimum oxygen zone (OMZ) already expend (see, Stramma et al works) and study on the shelf is of primary importance due to fishing activity which is particularly high in our study area and there is for instance report only from one sea survey cruise lead in 2013 (Ecoao within AwA project).

---

## Author Comment (AC2) · 10 Aug 2019

We are very grateful for the referee for his precious time and invaluable comments on the manuscript. The comments are encouraging and are useful for improving our manuscript. We have carefully provide the answers to all comments and the corresponding changes and refinements are reported below. While I believe the information presented is valuable and worth sharing, the manuscript has a number of serious issues which need to be addressed before publication. Some major points are as follows:

[Figure]

1. The authors do not identify the constituent animals of the SSLs. I realize this is often difficult or impossible when acoustic data are collected opportunistically, and that raw backscatter contains valuable information even when it cannot be attributed to specific animals or converted to biomass. However, it is extremely difficult to interpret these results without knowing at least broadly what kinds of animals are present. The authors need to provide more information on the likely sources of backscatter, even if it is only based on previous studies in this area. Answer : This is absolutely true, we do not present information on species composition, thus our study focus on ecosystem organisation (e.g., important for ecosystem modellers who usually and unfortunately do not consider species level communities) aiming to describe the spatio-temporal variation of SSLs. We agree that the lack of taxonomic information about the species composition of SSLs was a limitation. Taking into account these remarks, and as you requested, we provided more information on the specific compositions based on the literature in the area. The changes on the document are reported below: "The Petite Côte in the Senegalese coastal shelf is a nursery area for fish and is the main area in which juveniles of numerous species particularly small pelagic species concentrate (Diankha et al., 2018; Thiaw et al., 2017). This area is also known to be rich in zooplankton and micronekton. Many zooplankton groups are encountered over the Senegalese coastal shelf: : copepods, amphipods, annelids, appendicularians, chaetognaths, cirrhipeds, cladocerans, Decapoda, echinoderms, euphausiids, gasteropods, jellyfish, Mysidascea, ostracods, pelagic foraminifera, Protozoa, pteropods, Spumellaria.. Copepod is the most dominant group with a total abundance ranging from 50 to 90% (Anonymous, 2013; Ndour et al., 2018; Touré, 1971). Previous study (Tiedemann and Brehmer, 2017) on ichthyoplankton showed that Sparidae ($\sim$50%) was predominant followed by fewer Engraulidae ($\sim$8%) and Soleidae ($\sim$7%) while smaller proportions of Clupeidae and Carangidae ($\sim$4% each) as well as Myctophidae and Sciaenidae ($\sim$2% each)."

2. Some of the methods and results need to be described in more detail. In particular, the acoustic analysis and methods used to extract and measure the scattering layers are not sufiňAcient. The statistical models are also not described in enough detail. Answer : More details on the methodology of scatering layers extraction and statistical model are given below and added in the revised MS. In our study, the SSLs extraction was based on the use of a -75 dB threshold, which were indicated in the algorithm "contourf.m" of Matlab (see Matlab code at https://ch.mathworks.com/help/matlab/ref/contourf.html). This allowed to contour (by calculation of iso-lines according to the selected Sv threshold) the attached echo groups that formed the SSLs. The echoes within each SSL were extracted, and the associated descriptors were calculated from Matecho (open code already available on web see MS section 6 "software and code availability). The code "Layer" allows to recover echointegrated echogram and SSL descriptors calculated from Matecho, showed below and added on Appendix A in the revised version. The code "ComparEchoProfil" (Appendix B) allow to calculate mean Sv and sA profiles associated to each CTD profiles and to compare them (see below equations 1 to 3) and to extract CTD station meta information (see code below for details). We clarify inside the revised text of the MS: "After extracting SSLs with Matecho, we developed aMatlab extension of Matecho named "Layer" (Appendix A) to recover SSL thickness, minimum and maximum SSL depths (Dmin. and Dmax., respectively) and echointergrated echogram from Matecho output files and to provide it to another Matlab program "ComparEchoProfil" (Appendix B). ComparEchoProfil allows to fit in time and depth echointegrated echograms to the associated CTD vertical profiles. We used the equation to calculate thickness and also mean sA and Sv profiles based on the average of three ESUs: the ESU nearest to the CTD position and previous and following in correspondence with CTD depths (equations are not supported in this file, see the document pdf version added in supplement) ".

3. Some of the conclusions are not supported by the data presented. For instance, the authors refer to turbulence and advective transport, which were not measured. The lack of information on the sound-scattering species also makes parts of the discussion quitevague, reading more like abroad literature review rather than a specific discussion of the physical/biological processes likely to be at play in this particular instance. One finding which needs better explanation is the apparent reverse diel migration (up during day, down at night). One possibility which needs to be considered is cross-shore diel migration (see comment and suggested reference below). Answer : Indeed, this is true, nevertheless turbulence and advective transport are related to upwelling which is supported by our data. We consider in the discussion that these both processes (turbulence and advective transport) are more structuring than are the other basic studied parameters because responsible of the formation and/or dispersion of SSLs. For theses reason, we thought it was important to discuss them in this paper. But we recommend in future investigations to take speciafically into account these both parameters and partoculalry turbulence which is already plane in a sea survey which will be led in 2020 or 2021. We agree that lack of taxonomic information on the SSLs species composition was a limitation, that why we have tried to fill this gap with the literature review. As suggested, we made a focus on the reverse possible diel vertical migration found in the study area. These part have been improved in the document as detailed reported below: • Introduction " Zooplankton and micronecton are also known to undergo diel horizontal migration (DHM), moving them to within 1 km of the shoreline each night into waters shallower (Benoit-Bird et al., 2001). These DHM, like the DVM, that they often accompany, help organisms find food and avoid predators (White, 1998). " • Discussion " Diel variation was also observed for SSL acoustic density which showed opposite patterns in the two areas, i.e., higher up in the water column during night than day in the inshore area and higher up during days than at night in the offshore area. A possible explanation of this observed diel variation is the horizontal migration. DHM are known as nocturnal horizontal migration of both plankton and consumers into shallow and inshore waters (Benoit-Bird et al., 2001; Benoit-Bird and Au, 2006). DHM have been observed in marine copepods (Suh and Yu, 1996) which represent the main zooplankton group in the study area (Ndour et al., 2018; Rodrigues et al., 2017). It is hypothesized that these inshore–offshore migrations are a strategy for avoiding visual predators (White, 1998), and result in increased access to

food resources relative to simple vertical migration (Benoit-Bird et al., 2008)." • Conclusion " SSL acoustic density variation suggested different diel migrations : a normal and reverse DVM, and/or a DHM."

4. Several of the figures need revision (see specific comments at end). Answer : The outlined figures are revised (see minor comments answers) Minor comments, questions, and line edits follow below.

47: "entire mid-trophic level" Not necessarily. There can be a large "jelly web" that is not visible to echosounders, depending on their frequency. See e.g. Choy et al. (2017), https://doi.org/10.1098/rspb.2017.2116 72-89: This is a good overview of the literature, but I think the term "SSL" is used a little too broadly here. The authors should make it clearer that "SSL" is not a biological classification, and that animals making up SSLs can be all kinds of different things, with correspondingly different biological, physiological, and ecological needs. Answer : We modified this part by giving a new definition including large "jelly web" and adding Choy et al. (2017). We aggree that SSL is not a biological classification but include a wide variety of organisms. We defined it as a layer where a large number of pelagic organisms reflect the hydroacoustic signals, we define the SSL in the introduction as the sound reflected by zooplankton and micronekton organisms. We revised the definition of the SSLs and add some changes as reported below: "The SSLs represents a concentrated layer of marine organisms such as zooplankton aggregates and nekton that occur at specific depths (Benoit-Bird and Au, 2004; McManus et al., 2008) but also potentially jellyfish (Choy et al., 2017) which usually respond well at 38 kHz. Neverthless the "SSL" is not a biological classification, and that animals making up SSLs can be all kinds of different things, with correspondingly different biological, physiological, and ecological needs. The SSLs are dynamic, active, and have a particular behavior as a function of their community structure causing changes in their vertical distribution, size, and shape over time and space (Gómez-Gutiérrez et al., 1999). Zooplanktonic and micronektonic components are fundamental to ecosystem functioning, particularly in productive upwelling areas

(e.g., off the south coast of Senegal). "

90-91: Please clarify what is meant by "fine-scale" (m, 10s of m, etc.) Answer : Fine scale mean here 0.1 nmi * 1m depth. We clarify it in the document : "In this study, we use acoustic tools (Simmonds and MacLennan, 2005) to examine the fine-scale (0.1 nmi * 1m depth) vertical structure of SSLs (i.e., their depth in the water column, thickness, and density)".

106: The term "radial" in this context is not standardâËŸ AËĞTplease replace with "transect" or "transect line".

Answer : To standardize throughout the document we replaced radial by transect in all the document, tables and figures.

111: What parasite? Answer : Here we refer to noise parasite in echogram. It's defined as unwanted contributions to the signal from mechanical, electrical, or biological sources (Simmonds and MacLennan, 2005). We clarify it in the document : "Considering aft draught of the vessel, the acoustic near field and the presence of acoustics parasites (including air bubbles) in the upper part of the water column, we have applied an offset of 10 m."

112: "Offset" implies the data were shifted 10 m in space. I believe you mean that you discarded data from the upper 10 m? Answer : You are right, we mean that data upper than 10 m were discarded. In underwater acoustic, the term "offset", particularly the "surface offset" is a fixed depth derived from the location of the transducer below the water surface adding a zone for the transducer's near field. We have clarified it in the document : "Considering aft draught of the vessel, the acoustic near field and the presence of acoustics parasites (including air bubbles) in the upper part of the water column, we have applied an offset of 10 m (acoustic data above 10m have been deleted)."

121: Replace "extracted" with "removed." Answer : Absolutely right. Since we refer to SSLs extraction, we have rephrased it in the document as below : "After each echogram correction, we extracted the SSLs that were above the mean acoustic volume backscattering strength (Sv in dB) threshold of -75 dB (i.e., values below -75 dB were excluded from the analysis)." 122: A threshold of -70 dB at 38 kHz, as you say, is typical for fish surveys, but studies of smaller animals often use a threshold lower than -75 dB, which will eliminate backscatter from most zooplankton at this frequency. I'd like a little more explanation of how you selected this threshold. Answer : To choose the appropriate threshold value, we have varied the extraction thresholds between -80 and -65 dB and made visual inspections of echograms. We found that the -75 dB threshold appeared relevant to extract the SSLs (Figure revision 1). In addition, we have analyzed the sA variation according to the extraction thresholds, and noticed that below -75 dB the sA variation was weak (see below, a point of inflection is visible at -75dB);

Also, it appears from Figure 2 that the acoustic data were binned at coarser resolution than the raw data from the EK60. Is this the case? If so, what are the dimensions of the integrated cells? Answer : yes, the echos were integrated in cells of 0.1 nmi * 1m depth.

128: Replace "security" with "safety" Answer : Thank you. It's done in the document as reported here : "The inshore area is known to be rich in fish larva and eggs (Tiedemann and Brehmer, 2017) however, low sample number was collected in the coastal inshore water due to safety reason, i.e., research vessel investigate area> 20 m bottom depth. "

141: This link is no longer functional. Are these data available somewhere else? Answer : we provide here the good reference with DOI (JPL OurOcean: GHRSST Level 4 G1SST Global Foundation Sea Surface Temperature Analysis, , doi:10.5067/ghg1s-4fp01, 2010), and have corrected the paper as reported here: "Global High Resolution Sea Surface Temperature (GHRSST) data were extracted from daily outputs by the Regional Ocean Modeling System group at NASA's Jet Propulsion Laboratory (JPL

OurOcean, 2010). "

147-150: Please include a reference or hyperlink to the Matlab code in the body of the paper. Is there a reason it needs to be password protected instead of, say, posted publicly on GitHub? Even though the code is shared, the algorithm you used for detecting SSLs and extracting their descriptors still needs to be described in more detail. This is a non-trivial problem and different approaches can yield different results. Answer : We have included these matlab codes in the paper (in Appendix A, B) as suggested. Again, "Matecho" is an Open-Source Tool, it is simply the means used by the acoustics team of IRD (institute of research for sustainable development) to share their tools on the web. More details on the methodology of scatering layers detection and extraction are now given, see change in the MS below : To define the descriptors (Thickness, Depth, sA, Sv) of the SSLs, we used a threshold of -75dB, which were indicated in the algorithm "contourf.m" of Matlab (Matlab code can be found at https://ch.mathworks.com/help/matlab/ref/contourf.html ). This allowed contouring (by calculation of iso-lines according to the selected Sv threshold) the attached echo groups that formed the SSLs. The echoes within each SSL were extracted, and the associated descriptors were calculated from Matecho (code already available on world wide web see MS section 6 "software and code availability"). The pseudo code "Layer" ( Appendix A) allows to recover echointegrated echogram and SSL descriptors calculated from Matecho, showed below and added on Appendix A in the revised version. The pseudo code "ComparEchoProfil" ( Appendix B) allow to calculate mean Sv and sA profiles associated to each CTD profiles and to compare them (see below equations 1 to 3) and also to extract station meta information (see code below for details). We clarify inside the revised text of the MS as reported here: "After extracting SSLs with Matecho, we developed an ad hoc Matlab extension of Matecho named "Layer" to recover SSL thickness, minimum and maximum SSL depths (Dmin. and Dmax., respectively) and echointergrated echogram from Matecho output files and to provide it to another Matlab program "ComparEchoProfil". ComparEchoProfil allows to fit in time and depth echointegrated echograms to the associated CTD vertical profiles. We used
the equation to calculate thickness and also mean sA and Sv profiles based on the average of three ESUs: the ESU nearest to the CTD position and previous and following in correspondence with CTD depths (equations are not supported in this file, see the document pdf version added in supplement) ".

163-167: Move this explanation up to come before the preceding paragraph. The goal of an analysis should be described before the software used to accomplish it. Answer : yes, it is now done in the revised version of the MS, we have moved up the paragraph and replaced the paragraph related to software used at the end of this part as structured below : "We applied HCA to discriminate between water masses of inshore and offshore stations over the continental shelf based on CTD data collected at 10 m depth. We used PCA (Chessel et al., 2013) on the same dataset to determine similarities between CTD stations relative to environmental parameters. Physico-chemical parameters were standardized a-priori because they were measured with different metrics. Longitudinal variability (i.e., inshore - offshore area) of morphometric (thickness, depth) and acoustic characteristics (sA) of the SSLs are investigated in the discriminated groups considering bottom depth and diel period. Diel transition periods are removed from analyses to avoid SSL density changes bias due to diel vertical migrations. Transition periods are defined using sun altitude, i.e., around sunset and sunrise corresponding to a sun altitude between $\pm 18°$ (Lehodey et al., 2015). Morphometric and acoustic characteristics of the SSLs are also compared between the inshore area versus offshore area, and between day and night using student's t-test whose application conditions have been verified (normal distribution and variance equality). Echogram vs. profile coupling figures resulting from the "ComparEchoProfil" were analyzed to determine the relation between environmental parameters and SSLs. An ANCOVA test (analysis of covariance) (Wilcox, 2017) was implemented for SSLs characteristics (thickness, depth, and density). This ANCOVA model were developed on averaged data over station. The selection of the best model was performed using stepwise procedures. Stepwise selection was based on minimizing the Akaike Information Criteria (AIC) (Akaike, 1974). The relative importance of each variable in total deviance

explained was determined from the "relaimpo" R package (Tonidandel and LeBreton, 2011). Validity assumptions of the models was then assessed by checking for normality of distributed errors and homogeneity of residuals (Appendix B to C). For the ANCOVA, SSL density (sA) was log10 transformed for normality assumption. For all statistical tests, the significance threshold used was 0.05. We used R software (R Core Team, 2016) for statistical analyses and to map data. We used the R package 'Cluster' (Maechler et al., 2014) for Hierarchical Cluster Analyses (HCA) of CTD data, the R package 'maps' (Brownrigg, 2016) to map stations, the package 'ade4' (Chessel et al., 2013) to run a Principal Component Analysis (PCA), and the package 'oce' (Kelley, 2015) to display vertical section plots of physico-chemical parameters. "

168: Just say "Inshore-offshore variability." / 169: Use past-tense verbs, i.e. "were" instead of "are" Answer : yes, it's well noted and changed in the text : " Inshore - offshore variability of morphometric (thickness, depth) and acoustic characteristics (sA) of the SSLs were investigated in the discriminated groups considering bottom depth and diel period."

177-180: Please describe the statistical modeling in more detail here. I found it hard to figure out exactly what was being modeled/predicted and how. Answer : we have taken into account this remark and have improved this section as reported here : " Echogram vs. profile coupling figures resulting from the "ComparEchoProfil" were analyzed to determine the relation between environmental parameters and SSLs. ANCOVA tests (analysis of covariance) (Wilcox, 2017) were implemented for SSLs descriptors in each discriminated area (inshore and offshore). These models were set to predict each descriptors i.e., thickness, depth, and sA as function of temperature, density, DO, CHL, bottom depth and diel period. The ANCOVA models were developed on averaged data over CTD station."

209-210: "SST revealed and advection..." It is not clear to me how SST reveals advection here. Answer : we mean by this sentence that SST characterized by a well marked frontal zone isolating two water masses (warm waters and cold waters) showed an advection of water masses from offshore to inshore area. We understand that could be confusiong thus we delete this sentence (209-210).

211: Please make sure all verbs in this paragraph are past-tense. Also, this section might read better if it was organized by zone, rather than by variable. Answer : the paragraph organizations done due to the great inshore – offshore variability that exists and mostly variability between the three transects (longitudinal north – south). This choice makes it possible to do at the same time a detailed comparison of inshore – offshore and north-south variability and thus would like to keep it. We have taken into account the remark related to the verb tenses and have make all verb in past tense as written in the revised text : "At transect T1, a marked frontal zone appeared isolating two water masses between the 20 – 40 m isobaths (Fig. a1) which separated warm surface waters from deep cold upwelled water masses. At transect T2 and T3, the upwelling appeared as a cold-water tongue isolating a warm water band at the coast (Fig. a2, a3). At T3, this cold-water tongue was expanding toward the inshore area as well as to the offshore area (Fig. a3). Surface water masses of the inshore area were slightly denser than water masses in offshore area with approximately 26 kg m-3 and 25 kg m-3, respectively. For CHL, elevated concentrations were exclusively observed in the inshore area at radials R1 and R2. CHL was significantly higher in the inshore area than the offshore area with concentrations of 3.0 – 5.0 mg m-3 in the inshore area to 0.3 – 2.0 mg m-3 in the offshore area (Fig. c). At T3, the elevated CHL concentrations were observed in both inshore and offshore area close to the upwelling front. CHL was higher in the upper part of the water column (0 – 20 m) decreasing with depth in both areas. Higher DO concentrations were observed towards both sides of the upwelling core. At T1, the upwelling front was at the most coastal part separating the inshore area from the less oxygenated offshore area with DO concentrations of 5.0 – 7.0 ml l-1 and 4.0 – 5.0 ml l-1, respectively. At R2 and R3, the core moved towards the offshore, separating the inshore area (DO concentrations of 4.0 – 5.0 ml l-1) slightly more oxygenated than the offshore area (DO concentrations of 2.0 – 4.0 ml l-1). DO concentration decreased from the surface to bottom in both areas. "

216: I assume these are sigma-t values? Please clarify. Answer : It is sigma-theta [water density, kg/m-3]. We have specified that in the manuscript as written below: "At each station, sensors measured water temperature (°C), depth (m), fluorescence ($\mu$g l-1), water density, here sigma-theta (kg m-3), and dissolved oxygen (DO, ml l-1)."

233-234: I assume "depth" here refers to bottom depth? Some of the wording is a bit unclear, for instance "below 29 m depth." Does this mean "at locations seaward of the 29 m isobath," or "at locations where bottom depth was less than 29 m?" Answer : yes, her the depth refers to the botom depth. This mean that no SSL were observed at Radial R1, where bottom depth was below 29 m. We have corrected it in the revised paper : "Thickness and depth of the SSLs varied according to bottom depth in the inshore area and the offshore area. In the inshore area, on the northern transect T1, no SSLs were observed at coastal stations shallower than 29 m bottom depth (stations 1 and 2). In offshore stations, starting at 41 m bottom depth, the SSLs were observed in all stations and radials, and their thickness and depth increased with bottom depth. "

238-240: These two sentences are confusing to me, please clarify. If the difference was not significant, why report it? Answer : Indeed, we have first analysed sA variation of SSLs according to bottom depth, and found it's increase from inshore to offshore. We then tested if sA variation was significantly different between inshore and offshore area using Kruskal-Wallis test. The same test was used to compare SSLs thickness and depth between the inshore and offshore areas, and as expected, the results revealed theirs statistical difference between the inshore and offshore areas. However, as we have not reported the results for thickness and depth, we will remove this result in the paper as suggested by the referee.

242-255: This section reads like a string of numbers. It should be revised to empha-size concise verbal descriptions of the result, with the effect sizes and p-values either incorporated into the descriptions or reported parenthetically to support the verbal de-scriptions. Answer : we took into account the remarks of the referee and rewrote this

section focusing on the results. The changes are reported below: "The diel period had a significant effect on SSL thickness (p-value < 0.001), and SSL depth (p-value < 0.001) which were found both higher during the night in the inshore and the offshore areas. In the inshore area, during daytime, the mean depth and thickness of SSL were 19 and 11 m respectively, while during night, the mean depth and thickness were 46 and 35 m respectively. In the offshore area, SSLs were found at a mean depth and thickness of 49 and 38 m, respectively during daytime, while during night-time SSLs, depth and thickness were 86 and 75 m, respectively. Mean sA of SSLs varied also between day and night but were not significantly different (p-value = 0.890). In the in-shore area, the mean sA was 24 m2 nmi-2 during the day and 44 m2 nmi-2 during the night. In the offshore area, the mean sA was 46 m2 nmi-2 during daytime, and 25 m2 nmi-2 during night-time. "

269: "the model" there were actually several models, correct? This section needs more detail–the descriptions here and the tables to not include all the information referred to (for instance, the actual response variable and effect sizes at whose variance is being explained ?) Answer : It is right, we have three models for each area. The detail of all statistical results will be added in Appendix I, J, and K.

290-331: This section contains a lot of background information which would probably fił better in the introduction. Overall, I think it could be tightened and shortened con-siderably. Answer :as suggested, we have shortened this part by moving some parts pointed by the referee in the introduction. Find the changes made in the following text (and in MS with revision tracks): Introduction: "Senegalese coasts are characterized by a seasonal upwelling (in winter and late spring), mainly drifted by wind variability, topography, and density stratification (Estrade et al., 2008). During the upwelling sea-son, northerly trade winds induce a strong upwelling core south of Dakar (Ndoye et al., 2014; Roy, 1998). The upwelling core is located over the shelf, and SST is lowest on the coastal side of the shelf break, increasing in both offshore and coastal directions." Discussion: "Analyzing the spatial structure of SST helped to understand the upwelling

dynamics along the Petite Côte. The SST pattern, measured at the time of our survey, were in line with prior studies. During the upwelling season (in winter and late spring), a tongue of cold water over the shelf isolates a coastal band of warm water from the offshore area, and there is a surface divergence associated with the upwelling source over the shelf and convergence nearshore. The spatial difference of CHL concentration between the inshore area and the offshore area is the result of upwelled water carrying nutrients at the coast limited by water mass fronts. Nutrient-rich water, supplied to the sunlit surface layer by wind-driven upwelling stimulates the growth of phytoplankton that ultimately fuel diverse and productive marine ecosystems (Jacox et al., 2018). There is a link between the accumulation of biological material and the location of the coastal band of warm water. This coastal band between coast and the upwelling core has been regarded to function as retention area in which nutrient particles are trapped (Demarcq and Faure, 2000; Roy, 1998). The nutrient utilization is optimized by retentive physical mechanisms in the coastal area, which enhances microbial remineralization of particulate organic matter and zooplankton excretion, and then regenerates production through ammonium consumption (Auger et al., 2016). This causes an increase in primary production and results in a surplus of phytoplankton biomass in inshore areas. Low DO concentrations observed in the upwelling core separating more oxygenated water masses have been reported in previous studies (Capet et al., 2016; Teisson, 1983) over the Petite Côte. Once a water mass becomes isolated from the atmosphere, its oxygen content starts to decrease due to biological remineralisation of dissolved organic matter (Emerson et al., 2008; Machu et al., 2019). These low-oxygen bottom waters are transported to the inner shelf during upwelling favourable wind events. Moreover, temporal stability of the upwelling core is also noticeable over periods of several days to weeks; and export from the shelf to the open ocean is retarded (Capet et al., 2016). Thus, in such favorable condition of continuous food supply, photosynthesis may foster an enrichment of DO in the inshore. This is in line with high CHL levels observed towards both side of upwelling core, particularly in the inshore area. "
305: "surface divergence" Was this measured, or just inferred? Answer : it was not measured, it refers to the obvious barrier between both water masses (visible on Fig.1) induced by upwelled water mass. To avoid any incomprehension, we have replaced divergence per separation in the MS as reported: "A tongue of cold water over the shelf isolated a coastal band of warm water from the offshore area, and there was a surface separation associated with the upwelling source over the shelf and convergence nearshore." 333-351: I think some of these explanations need more consideration (or more explanation). Again, "SSL" is not a biological descriptor, and depending on the animals which are scattering the sound they may react in many different ways to physical forcings. Answer : we agree "SSL" is not a biological classification, and include a wide variety of organism as stated above. Indeed, in this paragraph we describe SSLs spatial distribution using some descriptors (thickness, depth and sA). It is obvious that, theses descriptors are related to SSLs biological composition and physical process that occur in the study area. However, our goal consists of studying SSLs spatial orgnisation over the shelf. As we clarified it above, our aim was to study SSL using descriptors, we did not investigate the individual behavior of the organisms constituting the SSLs although it can allow us to interpret our results that will be the subject of futher works. Here would like to remain generic.

338-339: What evidence is there that rapid advection causes low residence times? Are you sure the scattering animals are even planktonic? Answer : We based this assertion on previous studies revealing that the study area is rich in zooplankton (mostly copepod). But you are right that is not obvious that rapid advection causes low residence times (see e.g. Tiedemann and Brehmer 2017 paper who are co-author of this work). We have moderated the sentence adding "probable" in the former version. We can clarify adding "we can assume that" at the begining of the revised sentence.

343-344: Different animals can respond totally differently to different physical forcings. Urmy and Horne(2016) found that SSLs moved up and probably offshore during upwelling, but in the same ecosystem SSLs can actually intensify during

upwelling as krill and anchovies swarm more closely (Benoit-Bird et al. 2019, https://doi.org/10.1029/2018GL081603) Answer : We thank the referee for the interesting references and the explanations provided. We agree that animals can respond totally differently to different physical forcings. Referring to these papers, and as described it in our manuscript, marine organisms aggregate into SSLs or are dispersed depending on the upwelling intensity. Indeed, when upwelling was strong, both krill and anchovies were found in small, discrete aggregations, while during relaxation and reversals, forage biomass was more diffusely distributed, as evidenced by both acoustic and video observations (Benoit‐Bird et al., 2019). Urmy and Horne (2016) found that SSLs moved up and probably offshore during upwelling, and observed a decline in backscatter intensity in the upper water column immediately following an upwelling event. The new reference is added, and the changes in the paper are reported as written below : " Otherwise, different animals can respond totally differently to different physical forcings. Many authors have stressed that SSLs need stable hydrological conditions to form (Aoki and Inagaki, 1992; Baussant et al., 1992; Marchal et al., 1993). As example, in Monterey Bay (California), Urmy and Horne (2016) observed a decline in acoustic backscatter intensity in the upper part of the water column immediately following an upwelling event. In a more recent study, Benoit‐Bird et al (2019) found that when upwelling was strong, both krill and anchovies were found in small, discrete aggregations, while during upwelling relaxation and reversals, forage biomass was more diffusely distributed. Therefore, we assume that the increase of SSL thickness with depth from inshore to offshore off Senegal is caused by upwelled waters that disrupt the vertical stability of the water column."

351: Couldn't the SSLs be thicker and deeper farther offshore because there is more room in the water column? Answer : Of course, the SSLs are first constrained by the bottom depth i.e. availability of room. It's make sense that SSLs are deeper and thicker in the offshore area. Then to be clear we will precise in the text that : "Therefore, although the SSLs are first constrained by the bottom depth (i.e., room available), we assume that the increase of SSL thickness with depth from inshore to offshore off

Senegal is caused by upwelled waters that disrupt the vertical stability of the water column ".

364-365: One explanation for this phenomenon might be diel horizontal migration. See for instance: Benoit-Bird, K. J., Au, W. W. L., Brainard, R. E., and Lammers, M. O. 2001. Diel horizontal migration of the Hawaiian mesopelagic boundary community observed acoustically. Marine Ecology Progress Series, 217: 1–14. Answer : we thank the referee for the possible explanations and for the references provided to explain the possible reverse diel variation observed for SSL acoustic density. We agree that diel horizontal migration (DHM) of marine organisms constituting the SSLs is a plausible explanationeven in on the shelf we can suspect less amplitude than for mesopelagic fish. Therefore, this aspect of DHM will be added to the revised paper in the introduction, the discussion and in the conclusion as reported below: • Introduction " Zooplankton and micronecton are also known to undergo diel horizontal migration (DHM), moving them to within 1 km of the shoreline each night into waters shallower (Benoit-Bird et al., 2001). These DHM, like the DVM, that they often accompany, help organisms find food and avoid predators (White, 1998). " • Discussion " Diel variation was also observed for SSL acoustic density which showed opposite patterns in the two areas, i.e., higher up in the water column during night than day in the inshore area and higher up during days than at night in the offshore area. A possible explanation of this observed diel variation is the horizontal migration. DHM are known as nocturnal horizontal migration of both plankton and consumers into shallow and inshore waters (Benoit-Bird et al., 2001; Benoit-Bird and Au, 2006). DHM have been observed in marine copepods (Suh and Yu, 1996) which represent the main zooplankton group in the study area (Ndour et al., 2018; Rodrigues et al., 2017). It is hypothesized that these inshore–offshore migrations are a strategy for avoiding visual predators (White, 1998), and result in increased access to food resources relative to simple vertical migration (Benoit-Bird et al., 2008)." • Conclusion " SSL acoustic density variation suggested different diel migrations : a normal and reverse DVM, and/or a DHM."

[Figure]

378-379: Again, the term "SSL" is being used quite broadly here. Without some reference to the actual animals making up a particular SSL at even in broad taxonomic or trophic terms, it is hard to talk informatively about particular environmental inïñĆuences. Answer : As already meantioned in previous answer, we agree "SSL" is not a biological classification, and include a wide variety of organism as stated above. Indeed, in this paragraph we describe SSLs spatial distribution using our descriptors (thickness, depth and sA). It is obvious that, theses descriptors are related to SSLs biological composition and physical process that occur in the strudy area. However, Our study is primarily descriptive because it consists of studying SSLs spatial orgnisation over the shelf. As we clarified it above, our aim was to study SSL using descriptors, we did not investigate the individual behavior of the organisms constituting the SSLs although it can allow us to interpret our results. In the revised MS, we add in our case study in line 378-379, even if it is hardly presumed that main part of the SSL is formed by copepod.

392: If DO is relatively high everywhere, does it require this much discussion? Answer : in fact, local conditions of hypoxia exist in some stations. this was noticed by the first referee, and we thank him. We have corrected it in the manuscript (line 390): "Previous studies (Bertrand et al., 2010; Bianchi et al., 2013; Netburn and Koslow, 2015) have suggested that vertical distributions of SSLs organisms are limited by mid-water DO concentrations which constraint SSLs depth. These authors found a relationship between SSLs depths and hypoxia. However, in our study, we found correlation between SSLs (depth, thickness) and DO as expected, but vertical distribution of SSLs was not constrained by DO. Despite hypoxia local condition found in some stations (DO <1.42 ml l-1), SSLs appeared; Consequently, DO was not a limiting factor".

415: "primarily a function of temperature..." I would say light is at least as important as temperature, especially for migrating animals. Answer : it's absolutely true, as it's known as mentioned in the introduction that DVM are mainly influenced by environmental factor, mainly light, nutrient, and temperature and also by biological factors (feeding

and predator-prey interactions). We just wanted to discuss the factors used in our study. However, to correct we will simply write that: " DVM behaviors are influenced by environmental cues (e.g., light, nutrients, and temperature) and predator-prey interactions (Clark and Levy, 1988; Lampert, 1989). Relative changes in light intensity are identified as the most important proximate stimuli driving DVM, including the amplitude of the migration as well the timing of the up and downward movement (Meester, 2009). SSLs vertical distribution is known also to be a function of temperature (Bertrand et al., 2010; Hazen and Johnston, 2010; Netburn and Koslow, 2015). "

418-419: On the previous page, you said DO had only a limited influence on the SSLs' positions? Answer : right, in previous page we have written : "In our study, DO appeared to have a limited influence on SSLs vertical position, no doubt due to high DO value in both area." We mean in this previous sentence that SSLs vertical distribution was not constrained by DO. To avoid any confusion, we have changed this text by the following: "In our study, the results suggested that DO also influenced SSL depth and thickness. Despite local hypoxia noted in some stations, SSLs vertical distribution was not constrained by DO. "

445-446: The authors did not measure either turbulence or advection, so I don't think this assertion in the concluding paragraph is supported. Answer : as mentioned above, although turbulence and adjective transport has not been measured, we refer to these parameters in this paper because they are directly related to upwelling and are responsible of the formation and/or dispersion of SSLs. We will consider this remark in the conclusion and mention upwelling rather than turbulence and advection which are now more moderated. The changes are reported below: "SSLs were influenced by the strong resurgence in the upwelling (triggering at physical level turbulence and advection), which lead to change in SSLs structures (thickness, depth and sA)."

General comment for all heatmap-style figures: please redo these with perceptually uniform colormaps (such as the "parula" colormap which is now the default in Matlab). As-is, these figures will not reproduce well in black and white, and will be difficult

for color blind readers to interpret. Even in color, the un even luminance introduces arti-
facts which make interpretation more difficult. For instance, in Figure 1, temperatures
of 19.5, 20.5, and 22.5 all have similar brightness values, and the red-to-green color
ramp in the middle of the scale will be very difficult for red-green colorblind readers
to interpret. Answer : we have changed the figure taking into account the remarks (see
Figure revision 3) but in this new figure the contrast of the two zones is not highlighted.
So, we suggest to keep the first figure and to add this new one in Appendix.

Figure 2: How were the acoustic sections in the top panels selected? The first one
is more than 12 hours long, while the second is only 3 hours. It would be helpful on
both of these to show a distance scale, as well as periods of light and dark. Also, the
red lines indicating the station locations are difficult to see, especially in part (a).
Answer : the first panel has been selected over a distance of 1000 ESU (here 1 ESU
= 0.1nmi), while the zoomed echogram is a portion over three ESUs: the ESU nearest
to the CTD position (ESU ctd) and previous and following ESU in correspondence with
CTD depth. Following the comments of referee #1 ("For ComparEchoProfil, one 0.1
nmi unit is used for calculation around the CTD cast position, but in the figures several
units are shown (normally 5). Since these are not averaged in "ComparEchoProfil", so
I would leave that out so that the reader is not confused by the variability."), we have
removed the zoomed echogram (see below) to avoid any confusion. See below the
improved figure (Figure revision 4) taking into acccount all referee's remarks.

Figure 4: This figure is a repeat of Figure 1 and can be deleted. Answer : well
noted, we have deleted this figure in the revised version and add it in Appendix G.
Figure 5: It would be really valuable to add a fifth column with average backscatter
to this figure. Answer :Indeed, we have added a column with average sA (log10
transformed value) to this figure as presented in figure revion 5.

Figure 6: This figure is very difficult to read, and should be redone it in a different
format. I would suggest making three subplots, one for each transect ("radial"). Each
of these subplots would have bottom depth or distance offshore on the x-axis, and

depth below the surface on the y-axis. Mean layer depth at each station would then be plotted as a point, with the size of the point proportional to the layer's NASC. Error bars/whiskers above and below each point would give a visual indication of its thickness. Points could be colored differently to show if they were recorded in daytime or nighttime. Answer : we have also modified this figure as presented as presented in figure revion 6.

Please also note the supplement to this comment:
https://www.ocean-sci-discuss.net/os-2019-23/os-2019-23-AC2-supplement.pdf

[Figure]

**Figure 1 revision: [only for referee] ;** representation of an ECOAO echogram (from 06[th] -07[th]) at three frequencies. Each frequency (here 38, 70 and 120 kHz) was scaled from 0--255 and assigned a colour RGB value based on its frequency. This method allow to clearly observe that the signal is mainly associated to 38 kHz and allow to find the same SSL shape than combined frequency.

**Fig. 1.**

[Figure]

**Figure revison 1 [only for referee]: Variations in mean Nautical Area Scattering Coefficient (NASC or $s_A$) of sound scattering layers (SSLs) according to the volume backscattering strength ($S_v$) threshold for ECOAO acoustic sea survey data collected from 6–8 March 2013.**

**Fig. 2.**

[Figure]

Fig. 1: Location of the survey area off the southern Senegalese (West African) coast. The hydroacoustic survey was conducted with FRV Antea (IRD) from Dakar (Cap Vert peninsula in the north) to the northern border of Gambia (horizontal black line). CTD-probes collected data at stations along three transecttransects perpendicular to the coast (R1 to R3). Sea surface temperature (SST, °C) were averaged over the three days of CTD sampling from the 6–8 March 2013. Stations of Group 1 (blue circles) occurred in the inshore zone, whereas stations of Group 2 (red triangles) were situated more offshore.

**Fig. 3.**

[Figure]

Fig. 1: Echograms and associated vertical acoustic profiles as well as physico-chemical parameters (CTD data) for two example stations: (a) station 19 in the "inshore area" and (b) station 12 in the "offshore area". For both (a) and (b), top panels are echogram data collected along the transect (nmi), whereas the bottom panels depict acoustic and environmental data (depicted by the red vertical line in top panels). Environmental data for the sound scattering layer (SSL) collected at the stations at the time depicted by dotted vertical lines. Data represent mean conditions for the station collected within an area of 0.1 nmi area around the station: acoustic volume backscattering strength ($S_v$) SSL, temperature profile SSL, CHL profile SSL, oxygen profile SSL, and density profile SSL.

**Fig. 4.**

[Figure]

Fig. 5: Mean vertical profiles of (a) temperature, (b) density, (c) chlorophyll-a concentration, (d) dissolved oxygen, and (e) square rooted Nautical Area Scattering Coefficient (sA) in the three transects (T1, T2, T3; see Fig. 1) with positions of vertical probe stations CTD in the inshore area (vertical line in blue (G1)) and the offshore area (vertical line in red (G2)).

**Fig. 5.**

[Figure]

Fig. 6: Sound scattering layers (SSLs) mean depth (dot) according to bottom depth where they were detected, with their associated SSL thickness (line in meter), and SSL mean Nautical Area Scattering Coefficient (NASC or $s_A$ in $m^2$ $nmi^{-2}$), along transect T1 (north), T2 (intermediary), and T3 (south) during nighttime (in black) and daytime (in grey) sampling periods.

**Fig. 6.**

Appendix I : Result of ANCOVA models between thickness of sound scattering layers (SSLs) and environmental parameters (temperature, density, dissolved oxygen, chlorophyll-a, diel period and bottom depth) in the ishore area (G1) and the offshore area (G2). [G1: Multiple R-squared: 0.869, Adjusted R-squared: 0.8515, p-value: 0.000]; and [G2: Multiple R-squared: 0.8557, Adjusted R-squared: 0.7956, $p$-value: 0.000). Significant p-value in bold.

| | Coefficient Estimate | | Std. Error | | t value | | $p$-value | |
|---|---|---|---|---|---|---|---|---|
| Group | Inshore | Offshore | Inshore | Offshore | Inshore | Offshore | Inshore | Offshore |
| Intercept | -11.865 | 56030 | 4.161 | 17610 | -2.85 | 3.18 | **0.012** | **0.007** |
| Bottom depth | 0.916 | 0.21 | 0.148 | 0.06 | 6.15 | 3.41 | **0.000** | **0.005** |
| Diel period (Night) | 11.492 | 27.35 | 3.735 | 8.69 | 3.07 | 3.14 | **0.007** | **0.008** |
| Temperature | - | -383.80 | - | 119.60 | - | -3.21 | - | **0.007** |
| Density | - | -1898 | - | 598.50 | - | -3.17 | - | **0.008** |
| Oxygen | - | -1.76 | - | 0.55 | - | -3.19 | - | **0.007** |

Appendix J : Result of ANCOVA models between depth of scattering layers (SLs) and environmental parameters in the inshore area (G1) and the offshore area (G2). [G1: Multiple R-squared: 0.8056, Adjusted R-squared: 0.7797, p-value: 0.000]; and [G2: Multiple R-squared: 0.8557, Adjusted R-squared: 0.7956, p-value: 0,000]. Significant $p$-value in bold.

| | Coefficient Estimate | | Std. Error | | t value | | $p$-value | |
|---|---|---|---|---|---|---|---|---|
| Group | Inshore | Offshore | Inshore | Offshore | Inshore | Offshore | Inshore | Offshore |
| Intercept | -4.223 | 56040 | 5.603 | 17610 | -0.75 | 3.18 | 0.4627 | **0.007** |
| Bottom depth | 0.954 | 0.21 | 0.200 | 0.060 | 4.76 | 3.41 | **0.000** | **0.005** |
| Diel period (Night) | 12.864 | 27.35 | 5.030 | 8.690 | 2.55 | -3.14 | **0.021** | **0.008** |
| Temperature | - | -383.80 | - | 119.60 | - | -3.21 | - | **0.007** |
| Density | - | -1898 | - | 598.50 | - | -3.17 | - | **0.008** |
| Oxygen | - | -1.76 | - | 0.550 | - | -3.19 | - | **0.007** |

**Fig. 7.**

Appendix K : Result of ANCOVA models between sound scattering layers (SSLs) density (log $s_A$) and environmental parameters in the inshore area (G1) and the offshore area (G2). [G1: Multiple R-squared: 0.398, Adjusted R-squared: 0.3178, p-value: 0.022]; and [G2: Multiple R-squared: 0.3448, Adjusted R-squared: -0.01258, p-value: 0.490]. Significant *p*-value in bold.

| Group | Coefficient Estimate | | Std. Error | | t value | | *p*-value | |
|---|---|---|---|---|---|---|---|---|
| | Inshore | Offshore | Inshore | Offshore | Inshore | Offshore | Inshore | Offshore |
| Intercept | -9.007 | -489.2 | 5.784 | 405.7 | -1.55 | -1.20 | 0.140 | 0.253 |
| Bottom depth | 0.028 | -0.001 | 0.009 | 0.011 | 3.04 | -0.96 | **0.008** | 0.357 |
| Temperature | 0.578 | 3.207 | 0.350 | 2.78 | 1.65 | 1.15 | 0.119 | 0.273 |
| Diel period (Night) | - | -0.126 | - | 0.202 | - | -0.62 | - | 0.546 |
| Density | - | 1.673 | - | 13.78 | - | 1.21 | - | 0.250 |
| Oxygen | - | 0.016 | - | 0.010 | - | 1.48 | - | 0.166 |
| Chlorophyll a | - | 0.679 | - | 1.22 | - | 0.55 | - | 0.590 |

**Fig. 8.**